



# Characteristics of joint heavy precipitation and high sea level events on the Finnish coast in 1961-2020

Mika Rantanen[1], Kirsti Jylhä[1], Jani Särkkä[1], Jani Räihä[1], and Ulpu Leijala[1]

[1]Finnish Meteorological Institute, Helsinki, Finland

**Correspondence:** Mika Rantanen (mika.rantanen@fmi.fi)

**Abstract.** Simultaneous heavy precipitation with high sea level can lead to more severe flooding than if the hazards occur individually. In recent years, the significance of compound flooding has gained attention in many coastal areas, but studies on compound flooding in Finland are still lacking. In this paper, we investigate the co-occurrence of heavy precipitation and high sea level events (hereafter compound events) on the Finnish coast, because the simultaneous occurrence of these events can be considered as an indicator for compound flooding. We use daily sea level observations from tide gauges and gridded precipitation data to extract the days when both sea level and precipitation are extreme on the same day. Reanalysis data is used to derive composite maps of the compound events, and these maps are then qualitatively compared to the maps of non-compound events (i.e. when only one of the two variables is extreme alone). Finally, the dependence of the compound events on monthly indices of different atmospheric circulation patterns such as North Atlantic Oscillation is studied with correlation analysis. Our results show that the occurrence of the compound events is mostly controlled by sea level variability, with climatologically the most compound events taking place in autumn and winter. In addition to a large inter-annual variability of the compound events, statistically significant increasing trends were observed at some tide gauges, especially in the Bothnian Bay. The composite maps of meteorological variables indicate that the compound events are associated with strong extratropical cyclones which bring moisture and push the storm surge towards the coast. In average, the compound events are associated with stronger winds near the tide gauges than those events when only sea level or precipitation is extreme alone. Finally, we found that a negative phase of Scandinavian pattern (i.e. upper-level through over Scandinavia) is the most favorable atmospheric circulation pattern for the occurrence of compound events, in particular during autumn.

## 1 Introduction

The frequency of extreme weather events has traditionally been investigated for single hazard events. However, this can potentially underestimate the risk of a harmful weather event and its impacts, because climate drivers causing extreme weather events often interact and are dependent on each other, spatially or temporally. For example, wildfires, droughts and heatwaves are typically triggered by the same underlying atmospheric processes and occur usually simultaneously. The co-occurrence of anomalous weather events, often called compound weather events, is a result of a combination of two or more variables which are not necessarily extreme themselves (Seneviratne et al., 2012; Leonard et al., 2014; Zscheischler et al., 2020). In its sixth assessment report, the Intergovernmental Panel on Climate Change defines compound weather event as "The combination of



multiple drivers and/or hazards that contributes to societal and/or environmental risk" (Seneviratne et al., 2021; Zscheischler et al., 2018). Many socio-economically important regions such as North America and western Europe have been recently found to be global hotspots for the occurrence of compound events (Ridder et al., 2020).

The joint occurrence of heavy precipitation and high sea level is a fairly common compound event which happens from time
to time in coastal areas. Heavy precipitation together with high sea level induce much higher risk for flooding damages than if the hazards occur individually. The high sea level in compound events is typically caused by a storm surge, a weather-generated temporary rise in sea level. Landfalling tropical cyclones, such as Hurricane Harvey in 2017 (Valle-Levinson et al., 2020), are one of the most typical drivers for compound flooding in lower latitudes (Lian et al., 2013; Gori et al., 2020), but outside the tropics, atmospheric rivers (Ridder et al., 2018) or mid-latitude storms (e.g. Hendry et al., 2019; Bevacqua et al., 2019) can
also induce compound flooding.

Compound flooding can cause severe impacts in low-lying, densely populated cities in mid-latitudes (e.g. Wahl et al., 2015). Recent examples of compound flooding include the events occurred in Australia (Kumbier et al., 2018), the Netherlands (van den Hurk et al., 2015) and Italy (Bevacqua et al., 2017). Hendry et al. (2019) concluded that the flood risk around UK coasts has likely been underestimated due to the lack of consideration of compound flooding events. Ganguli and Merz (2019)
studied compound flooding in northwestern Europe and showed that extreme sea levels and stronger storms greatly amplify river (fluvial) flood hazards. Thus, they emphasized the importance of including the compounding effect of extreme sea levels in river flood risk management.

It has been reported that the number of compound flooding events has increased significantly over the past century in many of the US coastal cities (Wahl et al., 2015). Furthermore, climate modelling studies indicate that anthropogenic climate change
will increase the probability of compound flooding in the northern Europe, especially in the coasts of UK, the Netherlands and Germany (Bevacqua et al., 2019). Meteorological drivers of compound coastal flooding are projected to increase globally by more than 25 % by 2100 compared to present, and in mid-latitudes compound flooding could become 2.5 times more frequent (Bevacqua et al., 2020b). Thus, in order to understand the factors causing the projected future increase of coastal flooding, investigating the characteristics of meteorological drivers leading to coastal flooding in the present climate is of great
importance.

Finland is a country located on the northeastern coast of the Baltic Sea (Fig. 1). Due to its location relative to the most common storm tracks, the coastline is often exposed to mid-latitude storms coming from the west or southwest, and the consequent storm surges caused by them. On the Finnish coast, storm surges, seiches (standing waves within and between the Baltic Sea basins), tides and wind-induced waves govern the sea level changes at short timescales (from minutes to days),
while the water exchange via the Danish straits and mean sea level rise steer the sea level variations slowly (from weeks up to years and decades) (Leppäranta and Myrberg, 2009). Wind and air pressure variations dominate rapid sea level changes on the Finnish coast, and studies show that the short-term variability in sea level has increased in Finland over the 20th century (Johansson et al., 2001).

Probabilities of coastal floods (accounting for both short- and long-term sea level variations) are expected to increase in
the future particularly along the south coast of Finland (Pellikka et al., 2018). High sea levels on the Finnish coast result





typically from concurrent effect of multiple components. This was the case e.g. in January 2005 when record-high sea levels were measured in the Gulf of Finland: the mean sea level in the Baltic Sea was over half a meter above the average, when additionally, standing waves and powerful Storm Gudrun hit the Finnish south coast and piled-up water towards the shore in an unparalleled manner (Pellikka et al., 2018). During this event, record-high sea levels were experienced also on the Estonian side of the gulf (Suursaar et al., 2006; Tõnisson et al., 2008; Mäll et al., 2017). Extreme sea level events on the Finnish coast are possible also due to the joint effect of still water level and wind induced waves (Leijala et al., 2018), and due to meteorological tsunamis that have been investigated on the Finnish coast not until the recent decade (Pellikka et al., 2020).

Compound flooding outside the Baltic Sea is considered in many previous studies, focusing on joint occurrence of high sea-level and high river discharge (e.g. Ward et al., 2018; Hendry et al., 2019; Ganguli and Merz, 2019). The study by Bengtsson (2016) investigates the probabilities of combined high sea levels and large rains in Malmö, Sweden. The main result of the study was that it is extremely rare that large rains occur simultaneously with high sea levels. Thus, they concluded that when evaluating the risk of flooding in Malmö one should concentrate on the probabilities of high sea level or large rains independently rather than on the combination of the two. However, Bengtsson (2016) focused only on one place which is located on the coast of Danish straits, and thus, the conclusions of their results are not directly applicable to the Finnish coast.

In this study, our first objective is to characterize the climatology, variability, and possible trends of compounding high precipitation and high sea level events at Finnish tide gauges. In order to understand the factors behind the daily variability of these events, we analyse the synoptic patterns driving the events. Based on earlier work (Hendry et al., 2019), we hypothesize that the leading cause for compounding high precipitation and high sea level events are closely-passing extratropical cyclones. Finally, to provide explanation for the possible long-term trends of the compounding high precipitation and high sea level events, our third objective is to analyse which atmospheric circulation patterns promote the occurrence of compound events in a temporally longer scale.

The outline of this article is as follows. First, in Sect. 2, the datasets used in this study are introduced. Then, the methods are described in Sect. 3. The main findings of the study are presented in Sect. 4: a climatology and variability of compound events on the Finnish coast; the synoptic patterns behind the events; and the atmospheric circulation patterns promoting the events. The article is finalized by the discussion and conclusions in Sect. 4 and 5.

## 2 Data

This analysis was based on four main datasets: i) in-situ sea level observations, ii) gridded precipitation data, iii) reanalysis data and iv) indices of atmospheric circulation patterns. These datasets are described next.

### 2.1 Sea level observations

Sea level observations from nine Finnish tide gauges for 1961-2020 were obtained from Finnish Meteorological Institute (FMI, Fig. 1 and Table 1). In total there are 14 tide gauges on the Finnish coast, but only nine of them were used in this study. Six observations per day are available until 1971, and hourly observations since 1971. For more information on the sea level




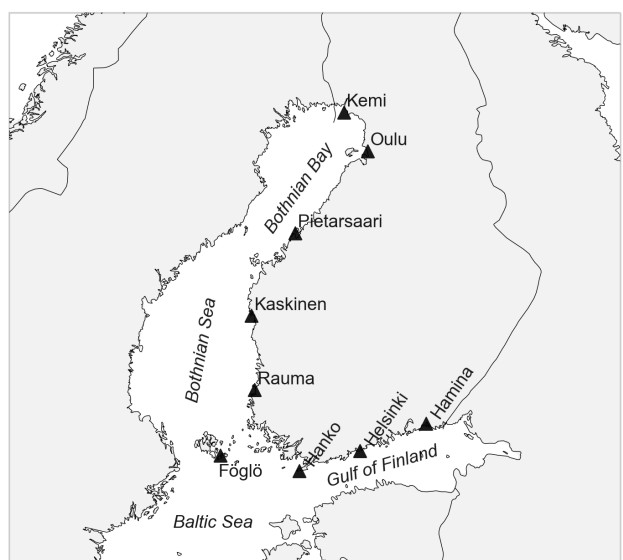

**Figure 1.** The locations of the tide gauges (marked with black triangles) used in this study.

observations in Finland, see Johansson et al. (2001). The vertical reference frame of the sea level observations used in this study was N2000 height reference system (Saaranen et al., 2009). We used one observation per day, defined as the highest hourly

reading which was observed between 06 UTC-06 UTC each day. This observation period was chosen as it was consistent with the precipitation observations which are recorded every day between 06 UTC-06 UTC (see section 2.2). To consider the decreasing trend of the relative sea level due to post-glacial uplift of the crust in Finland (Johansson et al., 2014), the sea level time series at each tide gauge were detrended. This was done by fitting a linear trend into annual mean sea level time series, and then removing this long-term trend from the daily sea level values. Thus, what we used for the analysis was the detrended

daily maximum sea level time series.

    Table 1 lists the tide gauges and the number of missing days of the sea level observations during the period of 1961-2020. Fortunately, the data is very uniform, and most of the tide gauges lack less than 1 % of data. The highest number of missing data can be found from Kemi tide gauge, in which about 2.5 years of data (968 days) during the 60-year study period are missing.

**2.2  Precipitation data**

For precipitation observations, we used daily gridded dataset called FMIClimGrid (Aalto et al., 2016). Among with several other meteorological variables, FMIClimGrid contains precipitation in a 10 km x 10 km grid, spanning from 1961 to present. FMIClimGrid is fully documented in Aalto et al. (2016), so it is only briefly described here. The precipitation observations used for the gridding are quality-controlled observations from the FMI database. Near the border regions, also observations from the

European Climate Assessment & Dataset (Klok and Klein Tank, 2009) database have been used. In total, 988 precipitation sta-



**Table 1.** Tide gauges used in the study, their coordinates on the Finnish coast, number of missing data in 1961-2020, and 95th and 98th percentiles of sea level and precipitation. See section 3.1 for details.

| Tide gauge | Coordinates (lat, lon) | Missing days | 95th sea level (cm) | 98th sea level (cm) | 95th precipitation (mm day$^{-1}$) | 98th precipitation (mm day$^{-1}$) |
|---|---|---|---|---|---|---|
| Kemi | 65.67, 24.52 | 968 (4.5 %) | 66 | 86 | 7.8 | 11.6 |
| Oulu | 65.04, 25.42 | 688 (3.2 %) | 65 | 86 | 6.9 | 10.3 |
| Pietarsaari | 63.71, 22.69 | 1 (0.0 %) | 55 | 70 | 6.7 | 10.4 |
| Kaskinen | 62.34, 21.21 | 70 (0.3 %) | 51 | 64 | 8.0 | 12.5 |
| Rauma | 61.13, 21.43 | 178 (0.8 %) | 49 | 62 | 7.7 | 11.5 |
| Föglö | 60.03, 20.38 | 762 (3.5 %) | 44 | 55 | 7.2 | 11.3 |
| Hanko | 59.82, 22.98 | 60 (0.3 %) | 49 | 62 | 9.0 | 13.3 |
| Helsinki | 60.15, 24.96 | 0 (0.0 %) | 55 | 70 | 8.7 | 13.1 |
| Hamina | 60.56, 27.18 | 142 (0.7 %) | 64 | 83 | 8.7 | 12.8 |

tions were used in the gridding, but the number of stations has decreased towards the 21th century due to the automation of the measurement protocol. The gridding has been done with an interpolation method called kriging (Matheron, 1963; Goovaerts, 1999). As external predictors for the kriging, the elevation of the station, its relative altitude, and its proximity to the nearest sea and lakes were used.

To obtain the daily precipitation time series at the tide gauges, we selected 70 km x 70 km square (7 x 7 grid cells) from the gridded data which was centred to the nearest grid point of each tide gauge. The daily precipitation was then defined to be the value averaged over the 70 km x 70 km area. However, as FMIClimGrid is a land surface dataset, and because tide gauges are located at the coastline of the Baltic Sea, the selected 70 km x 70 km squares centred to the tide gauges had variable number of missing data points over the sea. These missing points were neglected when calculating the area-average. In other words,
the precipitation was averaged over the land area inside the 4900 km$^2$ area. While the exact size of the averaging domain was fairly arbitrary, using an area larger than a few grid points was found to reduce the random noise which can emerge when using only the few closest grid points to the tide gauge.

## 2.3   ERA5 reanalysis

The synoptic situations of the compound events were studied with reanalysis data. We used ERA5 reanalysis (Hersbach et al.,
2020) which has been produced by the European Centre for Medium-Range Weather Forecasts. ERA5 covers the period from 1950 to present with one-hour temporal resolution and 31-km spatial grid spacing (T639 in spectral space). We used ERA5 single level data which were regridded to 0.5° horizontal resolution. Mean sea level pressure (MSLP), total column water (TCW) and 10-metre wind speed (WS10) fields valid at 18 UTC each day were used since this is the time in the middle of the 06-06 UTC observation period used in the daily observations and thus represents best the synoptic pattern on those days.





## 2.4 Atmospheric circulation patterns

In total six different atmospheric circulation patterns (ACPs) were evaluated in relation to the number of compound events at the tide gauges. The ACPs chosen for this study were North Atlantic Oscillation (NAO), Arctic Oscillation (AO), Scandinavian (SCA), East Atlantic/West Russia (EA/WR), East Atlantic (EA), and Polar/Eurasian (POL) patterns. More information on the patterns are available in Climate Prediction Centre (CPC) of National Oceanic and Atmospheric Administration (NOAA) database (CPC, 2021). For this study, we used standardized monthly mean values of the pattern indices for the period of 1961-2020. These values are available from CPC database (CPC, 2021).

NAO, often considered also the leading mode of the variability in the Euro-Atlantic region (e.g. Hurrell, 1995), describes generally the pressure difference between the Azores high and Iceland low. NAO is related to the strength and position of the North Atlantic eddy driven jet stream, and has a strong impact on the weather and climate in Finland, in particular in the winter season.

SCA is sometimes considered as the second mode of the variability in the Euro-Atlantic region, and has a pattern which is orthogonal to NAO (i.e. NAO pattern is zonal and SCA pattern is meridional). In its positive phase SCA features a high pressure system/blocking high over Scandinavia, while negative SCA are characterized by low pressure system/troughing over Scandinavia.

EA, EA/WR and POL patterns are less commonly used and they describe the weather variability in Finland to a smaller degree than NAO and SCA. EA is structurally similar to NAO, but the pattern is located further south and therefore EA affects the weather more in western Europe than in northern Europe. Likewise, EA/WR is structurally similar to SCA, but is located further west. In its positive phase the blocking is situated over the British Isles and the trough is over western Russia. POL pattern describes the circulation in the whole circumpolar vortex. In its positive phase the whole circumpolar vortex is enhanced, and northern Eurasia and Siberia typically experience warmer than normal weather. The negative phase of POL reflects a weaker than average tropospheric polar vortex.

Finally, AO pattern is the leading mode of variability in the northern hemisphere, and captures the maximum amount of explained variance. Thus, AO is very similar to NAO, which explains most of the variance in the Euro-Atlantic region. During positive AO the tropospheric polar vortex is strong with low pressure area in the Arctic region, surrounded by high pressure areas at mid-latitudes. During negative AO, there is typically high-latitude blocking present in the Arctic. AO correlates greatly with NAO because their associated loading patterns are fairly similar.

## 3 Methods

### 3.1 Joint occurrence of extreme sea level and precipitation

We extracted days when the daily precipitation sum and daily maximum sea level exceed the predefined percentile thresholds at each tide gauge. For the thresholds, two levels of extremeness were considered. The first level, hereafter called as elevated level (EL), was defined to be the 95th percentile of the daily time series, and the second level, hereafter called as high level (HL)



**Table 2.** Percentile values used in the event types. For example, EL compound event are defined as days when both daily sea level maximum and daily precipitation at the tide gauge is greater than or equal to its 95th percentile.

| Event type | Sea level | Precipitation |
|---|---|---|
| Elevated-level (EL) compound | $\geq 95$ | $\geq 95$ |
| High-level (HL) compound | $\geq 98$ | $\geq 98$ |
| Non-compound sea level | $\geq 98$ | $\leq 95$ |
| Non-compound precipitation | $\leq 95$ | $\geq 98$ |

was defined using 98th percentile of the daily time series. Isolating events with two levels of extremeness was a compromise: on one hand, we wanted to select a threshold low enough to have a sufficient number of events to allow a reliable statistical analysis. On the other hand, selecting too low threshold would include compound events with only moderate impacts which
are less relevant for this analysis.

When calculating the numerical values corresponding to the percentiles of precipitation, also days with no precipitation were taken into account. Both EL and HL thresholds were defined separately for each tide gauge, and the actual numerical values corresponding to the thresholds range from 44 (55) to 66 (86) cm for EL (HL) sea level and 6.7 (10.3) to 9.0 (13.3 ) mm/day for EL (HL) 70 km x 70 km spatial average precipitation amounts (Table 1).

The joint occurrence of the events were obtained by simply counting the number of times when both precipitation and maximum sea level exceed the percentile thresholds on the same day. In other words, EL compound events were defined as those days when both precipitation and maximum sea level exceed the elevated (95th percentile), and HL compound events were defined as those days when both precipitation and maximum sea level exceed the high (98th percentile) level. These levels are listed in Table 2 and illustrated in Fig. 2 with black dashed (EL) and red solid (HL) lines. Note that the group of
EL compound events include also the group of HL compound events. We only considered the cases when the lag between the precipitation and sea level events was zero days, so the events must occur on the same day.

The compound events were compared to the days when either sea level or precipitation were extreme alone. These days were defined as days when one of the two variables (precipitation or sea level) reaches high level (98th percentile) while the other variable stays below elevated (95th percentile) level (see Table 2). Hereafter, these are called non-compound sea level or
non-compound precipitation events.

In the cases when the compound events took place on consecutive calendar days, the day with highest sea level was only included in the analysis. This practise was conducted in order to make the time series consist of as independent events as possible. The same practise was applied to non-compound sea level events, and for non-compound precipitation events, the day with the highest precipitation amount was only included.

Statistically significant trends of the events were detected using nonparametric Mann-Kendall test (Mann, 1945; Kendall, 1948), and their magnitude was determined using Theil-Sen's slope estimator (Theil, 1950; Sen, 1968). A python package called pyMannKendall (Hussain and Mahmud, 2019) was used to calculate these statistics.

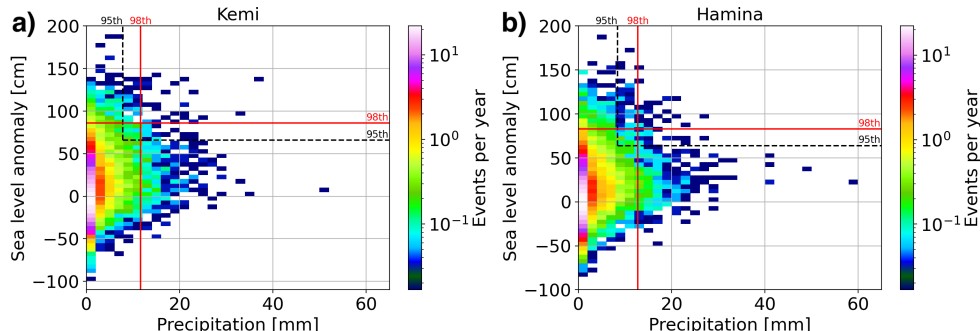

**Figure 2.** Two-dimensional density distribution of daily precipitation and daily maximum sea level at a) Kemi tide gauge and b) Hanko tide gauge in 1961-2020. The color depicts the number of events per year in each bin. Solid red lines show the boundaries of high-level (HL, 98th percentile) compound events and dashed line the boundary of elevated-level (EL, 95th percentile) compound events.

### 3.2 Composites of meteorological fields

Our second objective was to investigate the synoptic patterns which typically lead to compound events and analyse qualitatively
how they differ from those leading to non-compound events. For each tide gauge, we extracted the fields of mean sea level pressure (MSLP), total column water (TCW), and 10-metre wind speed (WS10) from ERA5 reanalysis on those days when HL compound and non-compound events took place. The fields are valid at 18 UTC each day, which represent the middle point of the observation period (06 UTC-06 UTC). Then, we derive composite plots of these fields (MSLP, TCW and WS10) by simply calculating the average of the fields. Thus, the composite plots represent the average synoptic conditions on those days when
compound and non-compound events have occurred. For compound events, we present here the composite plots only for the HL events, because they were more extreme and thus the average synoptic patterns are more contrasting to the non-compound events than in the case of EL compound events. Since the non-compound sea level and precipitation events are typically more frequent than HL compound events, they expectably have somewhat smoother composite maps.

### 3.3 Correlation to atmospheric circulation patterns

To link the interannual and decadal variability of compound events to large-scale atmospheric circulation patterns, Spearman's rank correlation (e.g. Wilks, 2011) was used to to calculate the correlations coefficients. Spearman's correlation was selected instead of the Pearson's correlation because Spearman's correlation assumes only monotonic relationship between the variables and is thus suitable for investigation of linkages with potential non-linear relationships. The relationship between ACPs and compound events were only conducted for EL events, because HL events are too rare for a reliable analysis. First, the number
of EL compound events and the average ACP indices in autumn (September-November), winter (December-February) and cold season (September-March) in each year were calculated. Then, the time series of the number of compound events and average ACP indices were detrended, and after that, their correlation coefficients were calculated.

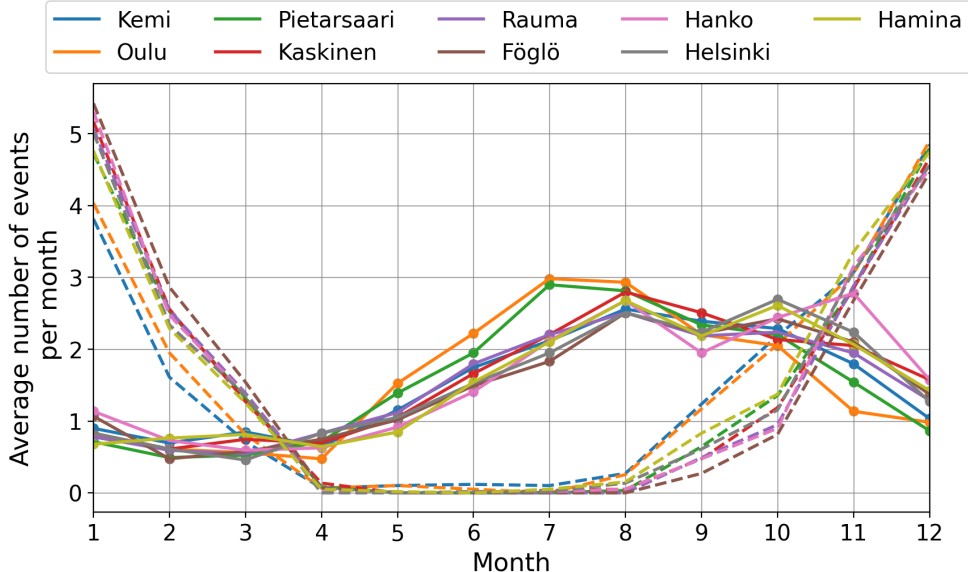

**Figure 3.** The monthly distribution of average number of extreme precipitation (95th percentile, solid lines) and sea level (95th percentile, dashed lines) events at the nine tide gauges on the Finnish coast. The numbers have been averaged over 1961-2020.

## 4 Results

### 4.1 Climatology of single events

Before discussing the characteristics of joint occurrence of extreme precipitation and extreme sea level (i.e., compound events), we present the seasonal cycles of these variables separately (i.e., single events) at the nine tide gauges (see their locations from Fig. 1). These events are defined as days when the precipitation or sea level exceed the 95th percentile threshold (see Table 1). The average numbers of these events per month are presented in Fig. 3.

The seasonal behaviour of the variables are certainly different. Extreme precipitation events (solid lines in Fig. 3) are most 215 frequent during the summer and autumn with a broad peak of occurrence taking place around August. In winter and spring extreme precipitation is clearly rarer, but not totally unprecedented. Compared to the extreme precipitation events, extreme sea level events (dashed lines in Fig. 3) have a sharper seasonal cycle. They are virtually absent in April-August but occur 2-3 times per month during winter, with January being clearly the peak month of occurrence.

In conclusion, Fig. 3 demonstrate that extreme precipitation and extreme sea levels tend to occur at different times of the 220 year: precipitation is most abundant in late summer when the atmosphere is warmest and its moisture content is highest. Instead, sea level variability and thus the highest sea level values are typically recorded in late autumn and winter when the short-term factors controlling the sea level variability such as storm surges, seiche oscillations, and wind-induced waves are at their most frequent. Due to the differences in the underlying physical mechanisms causing the events, the correlation between





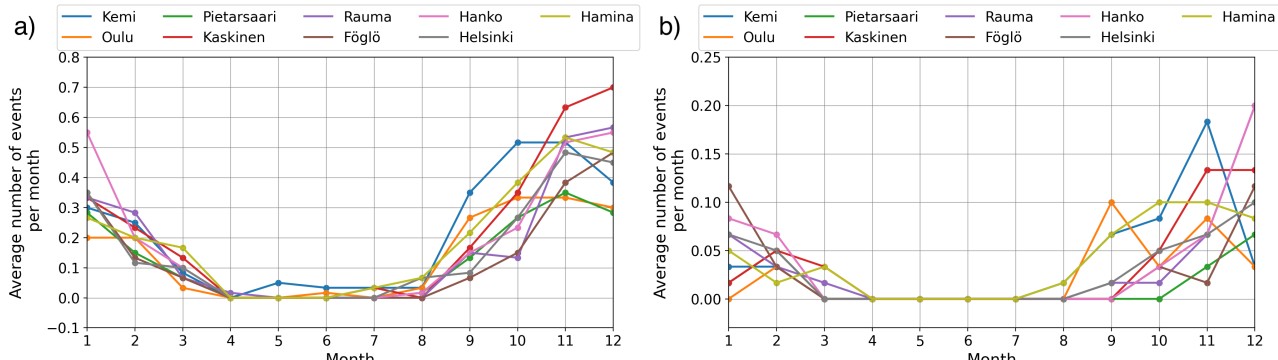

**Figure 4.** The monthly distribution of average number of a) EL and b) HL compound events at the nine tide gauges on the Finnish coast. The numbers have been averaged over 1961-2020. Note the different scales on y-axes.

precipitation and sea level is only modest. This is illustrated by Fig. 2 which shows the relationship of the events. The Pearson

correlation coefficient between the precipitation and sea level is 0.28 ($p$ <0.01) in Kemi (Fig. 2a) and 0.26 ($p$ <0.01) in Hamina (Fig. 2b). At the other tide gauge locations, the correlation coefficients ranged from 0.17 to 0.28 ($p$ <0.01).

### 4.2  Climatology, variability, and long-term trends of the compound events

The seasonal cycles of extreme precipitation and sea level overlap the most in autumn and early winter, so that is the time of the year when the compound events tend to occur (Fig. 4). EL compound events (Fig. 4a), which were defined using the 95th

percentiles of the variables, are naturally more common than HL compound events (Fig. 4b) which in turn were defined using the 98th percentiles.

In Kemi, HL compound events (Fig. 4b, blue line) account for 6 % of all days when sea level reaches the 98th percentile, and the same is true for all those days when precipitation reaches the 98th percentile (see the upper right quadrant of the red lines in Fig. 2a). In Hamina (Fig. 2b), the corresponding proportions are 6 % (sea level) and 7 % (precipitation). In other tide

gauges, the proportions ranged from 2 % to 6 %. Therefore, on the Finnish coast, extreme sea level and extreme precipitation rarely occur together, and it is more likely that only one of them is extreme alone than that they are both extreme at the same time.

EL compound events (Fig. 4a) do not generally take place in spring and summer due to absence of sea level extremes (Fig. 3). The peak month of occurrence in EL compound events varies between the tide gauges. Tide gauges located in the end of

the gulfs (Kemi, Oulu and Pietarsaari in the Gulf of Bothnia, and Helsinki and Hamina in the Gulf of Finland) experience their climatological peak earlier, in October or November, while the tide gauges located closer to the main basin of Baltic Sea (Kaskinen, Rauma, Föglö and Hanko) have their peak later in winter, in December or January.





The annual number of EL compound events at each tide gauge are shown in Fig. 5. The order of tide gauges in the figure goes from the northernmost tide gauge in the Bothnian Bay (Kemi, Fig. 5a) to the easternmost tide gauge in the Gulf of Finland
(Hamina, Fig. 5i). See Fig. 1 for their locations.

In average, the tide gauges in Finland experience 2-3 EL compound events per year. However, the time series of EL compound events shows substantial year-to-year variability. In some years, there are zero or a very few compound events while some years see more than 10 events. The annual number of compound events is naturally dependent on the percentile threshold used for defining the events. In Fig. 5, EL events (95th percentile) are shown.

The 60-year time series show also decadal variability in the compound events. 1960's and early 1970's are characterized by rather weak compound event activity. After 1970's, the number of events increased and most of the tide gauges show a peak in the 10-year running mean in 1980's (Fig. 5, orange line), after which the occurrence of compound events tend to decrease until late 1990's. The second peak, shown by the 10-year running mean in Fig. 5, occurs in late 2000's. Again, the mid-2010's saw lower compound event activity in many of the tide gauges after which the latest year, 2020, was one of the most active years
at the tide gauges. The decadal variability is expected to result from variability in large-scale climate indices, such as NAO or AO. We discuss this further in Section 4.4.

In addition to the decadal variability, some tide gauges exhibit statistically significant increasing trends of the compound events. These are Kemi (Fig. 5a), Oulu (Fig. 5b) and Rauma (Fig. 5e). When considering the trends in the EL precipitation events (solid lines in Fig. 3), the trends are increasing at all tide gauges (Fig. A1 in Appendix), in line with a previous study
investigating long-term trends of precipitation in Finland (Irannezhad et al., 2017). For EL sea level events (dashed lines in Fig. 3), only Kemi and Oulu have increasing trends (Fig. A2). Hence, the increasing trends of compound events (Fig. 5a and b) result from the combination of increasing extreme precipitation events and increasing extreme sea level events in recent decades and can be potentially linked to decadal changes in large-scale circulation patterns (Section 4.4)

In summary, the occurrence of compound events indicate substantial year-to-year variation, which is typical for the Finnish
climate and results from large natural variability of underlying weather patterns causing these events. In addition, the time series display decadal variability, with peaks in the 1980's and 2000's and valleys in the 1970's, 1990's and 2010's. However, these features are not clearly visible at all tide gauges and thus the detection of the underlying reasons is difficult.

### 4.3 Synoptic situations driving the events

This section investigates the synoptic situations which drive the compound and non-compound events on the Finnish coast.
To this end, we have calculated composite maps of mean sea level pressure (MSLP), total column water (TCW) and 10-metre wind speed (WS10) fields over the days when compound and non-compound events have taken place at the tide gauges. The composite plots were derived for all tide gauges, but here we focus only on two sites, Kemi and Hamina, as they are located in the different ends of Finnish coastline and provide the most contrasting composites.

The composite maps for Kemi tide gauge are displayed in Fig. 6. All three event types show cyclonic pattern over northern
Europe, but some essential differences between the composites exist. The compound events in Kemi (Fig. 6a and d) are typically driven by low-pressure system located off the Norwegian coast. A trough-like feature or in some cases a secondary

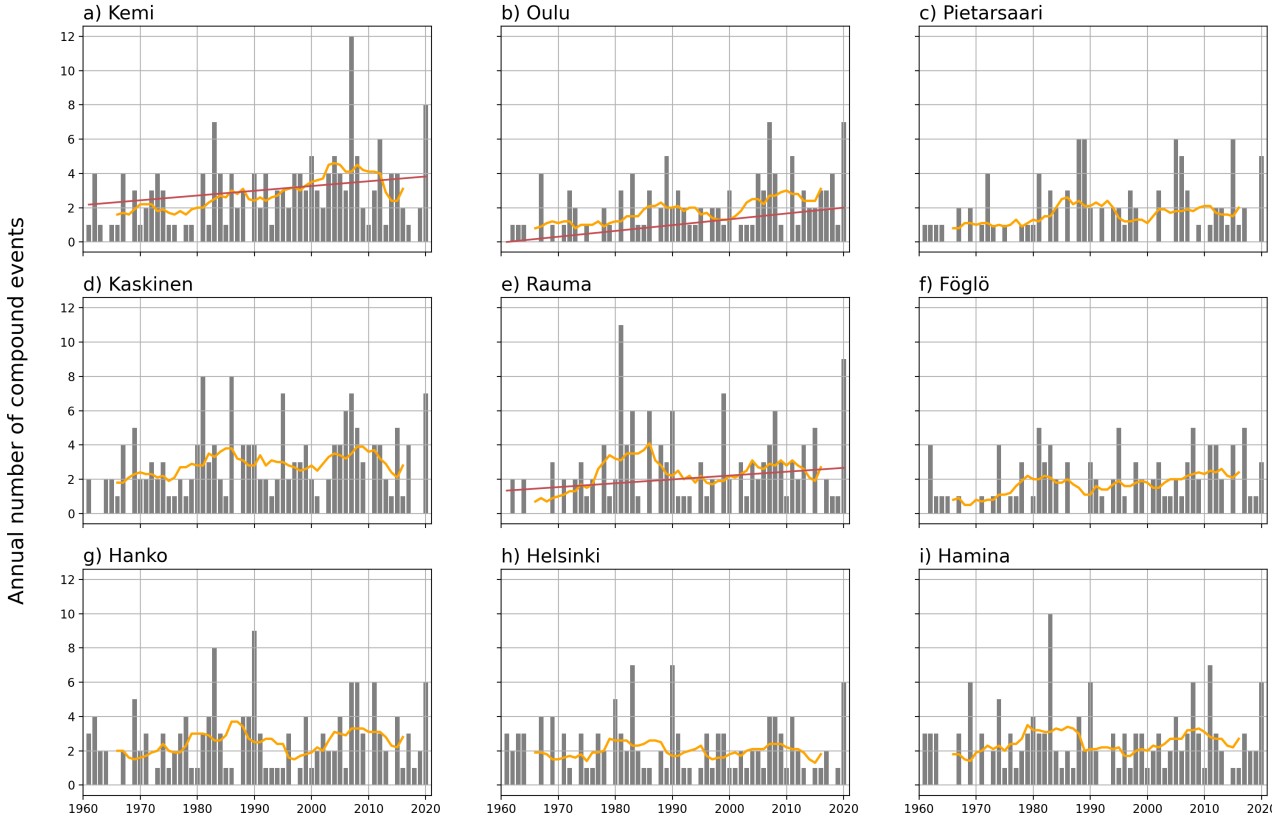

**Figure 5.** Annual number of compound events at the tide gauges (grey bars), 10-year running mean of the annual values (orange line) and the estimates of their linear trends calculated with Theil-Sen method (red line). The linear trends are plotted only for those tide gauges whose Theil-Sen trend estimate was non-zero.

low is evident over northern Sweden. Consequently, the pressure gradient on the southeast side of the trough is strong, and on average, WS10 values are up to 13 m s$^{-1}$ over the Bothnian Bay (Fig. 6d). The wind speeds are high also in the Bothnian Sea and Northern Baltic Proper on the compound event days, which allows the winds to push a surge into the end of the Gulf of

Bothnia and thus enable the extreme sea levels at Kemi tide gauge during the compound events.

Associated with the low-pressure system, a filament of high TCW extends from western Europe all the way to the Finnish Lapland (Fig. 6a). This filament has some characteristics of atmospheric rivers, which, according to Ralph et al. (2017), are defined as long, narrow, and transient corridors of anomalously strong horizontal water vapor transport.

The meteorological pattern on non-compound sea level events (Fig. 6b and e) is somewhat similar to compound events, but

the low-pressure system is typically situated further north, and MSLP values over southern Europe are higher. The zonal flow, indicated by MSLP gradient and WS10 (Fig. 6e), is strong over northern Europe. TCW values over Finland are markedly lower than during the compound events, so the feature resembling atmospheric rivers is absent. Wind speeds over the Gulf of Bothnia are generally weaker (Fig. 6e) than those during the compound events, which indicates that non-compound sea level events are

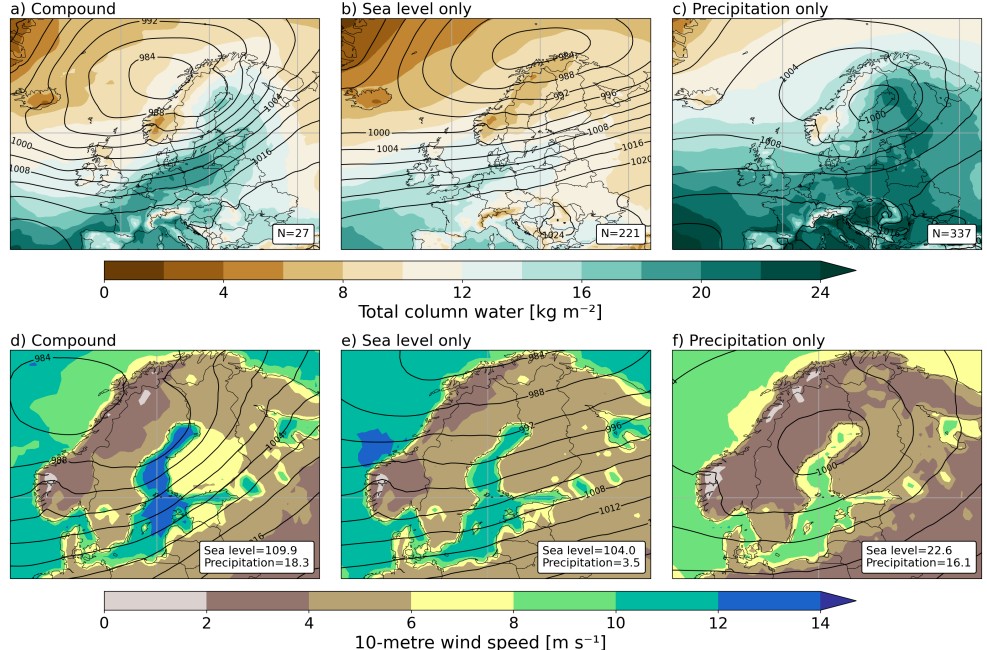

**Figure 6.** Composite maps of sea level pressure (hPa, black contours), atmospheric total column water (upper row, colours) and 10-metre wind speed (bottom row, colours) fields computed over compound events (a,d), non-compound sea level events (b,e) and non-compound precipitation events (c,f) at Kemi tide gauge. The numbers in lower right corners in the top row indicates the number of days used for calculating the composites. In the bottom row, the numbers indicate the average sea level height and precipitation during the events.

not driven by closely-passing storms but rather cyclones which pass Finland further north, on the Norwegian Sea and Barents
Sea.

Finally, the non-compound precipitation events in Kemi tend to be caused by weak low-pressure systems over the Bothnian Bay (Fig. 6c and f). Extreme precipitation events are the most frequent in warm season (Fig. 3), which means that also TCW values are generally higher than during compound (Fig. 6a) and sea-level events (Fig. 6b). The wind speeds near Kemi are generally weak (Fig. 6f), as expected.

At Hamina (Fig. 7), the synoptic situations leading to compound events are characterized by a low-pressure system traversing central Finland (Fig. 7a and d), to the north of the tide gauge location. The warm sector of the composite low-pressure system, as indicated by the higher TCW values, is located over eastern Europe and extends southern Finland. The MSLP gradient on the southern side of the low-pressure system is tight, and consequently strong westerly winds take place in the southern Baltic Sea and on the coasts of Baltic countries (Fig. 7d). The wind speeds in the Gulf of Finland are somewhat weaker than in the
main basin of the Baltic Sea.

The large-scale MSLP pattern on non-compound sea level events in Hamina (Fig. 7b) is somewhat similar to compound events (Fig. 7a), but there is no distinct centre of low-pressure system over southern Finland in the composite. Thus, and similar




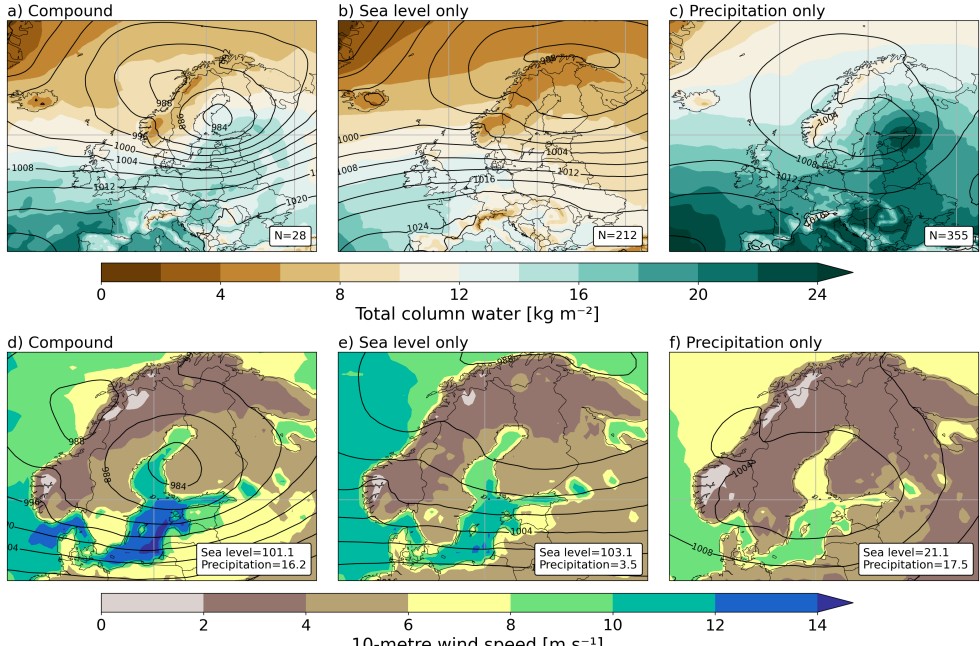

**Figure 7.** Same as 6, but for Hamina tide gauge.

to Kemi tide gauge, while compound events are mostly driven by a close passage of extratropical cyclone, non-compound sea level events are driven, on average, by cyclones passing the tide gauge further away. Accordingly, the TCW values are lower in eastern Europe (Fig. 7b), and average wind speeds over the Baltic Sea are notably lower (Fig. 7e) in the non-compound sea level events than during the compound events (Fig. 7d).

Interestingly, despite the stronger wind speeds associated with the compound events, the average sea level height on compound events (101.1 cm, Fig. 7d) is slightly lower than during the non-compound sea level events (103.1 cm, Fig. 7e). However, according to two-sided t-test, the difference is not statistically significant. In addition, the wind speeds are notably higher only in the southern Baltic Sea and not in the Gulf of Finland, and, moreover, the wind speed is not the only factor which affects sea level but also seiche oscillations and total water volume in the Baltic Sea play a role (see Introduction)

Consistently with Kemi (Fig. 6c), the non-compound precipitation events in Hamina (Fig. 7c) occur mostly in warm season (not shown) and thus the circulation field and wind speeds are rather weak (Fig. 7c and f). TCW values are locally high near Hamina, indicating the potential for extreme precipitation amounts.

In conclusion, the compound events at the tide gauges are associated with the close passages of extratropical cyclones which transport the moisture from lower latitudes and hence create the potential for high precipitation amounts. In this paper, we presented the composite maps for Kemi and Hamina, but the same conclusion applies also for other tide gauges (not shown). The strong winds associated with these systems push the storm surge towards the tide gauge and cause simultaneous risk for flooding via high sea levels.




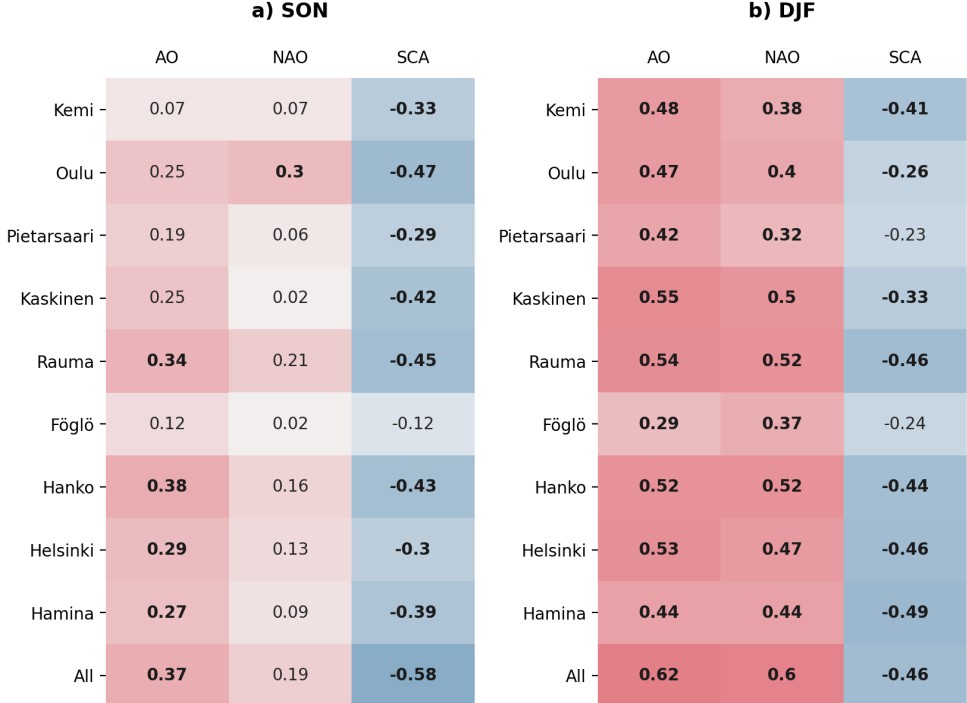

**Figure 8.** Spearman rank correlation coefficients ($r$) between the number of compound events and the average atmospheric circulation patterns indices in autumn (left) and winter (right). Reddish (blueish) colour indicates positive (negative) correlation. Statistically significant correlations (p < 0.05) are given in bold. See Section 2.4 for the abbreviations. "All" on the last row means the total number of compound events, summed up over all tide gauges.

## 4.4 Atmospheric circulation patterns

As evidenced in the previous section, the compound events at the tide gauges are associated with the passage of extratropical cyclones. Thus, the occurrence of compound events can be potentially linked to large-scale atmospheric circulation patterns (ACPs) that favour the passage of extratropical cyclones. In this section we investigate which circulation patterns promote the occurrence of EL compound events at the tide gauges. We first consider the average ACP indices over autumn (September-November) and winter (December-February), because compound events generally occur on these seasons (Fig. 4).

When considering the total number of EL compound events summed across all the nine tide gauges in autumn, the most influential ACP is Scandinavian pattern (SCA) with a correlation coefficient ($r$) of -0.58 ($p$ <0.05, Fig. 8a, bottom row). Negative SCA features a large upper-level low pressure over Scandinavia, which is favorable for the activity of extratropical cyclones (ETCs) and hence the reason why compound events correlate negatively with the SCA index. For autumn compound events at the individual tide gauges, SCA correlates significantly with almost all of them, and with consistently higher correlation coefficients than Arctic Oscillation (AO) and North Atlantic Oscillation (NAO) (Fig. 8a).





In winter, the influence of SCA is smaller and AO becomes the dominant pattern ($p < 0.05$, $r = 0.62$, Fig. 8b, bottom row). Positive AO (and NAO) reflects a zonal pattern which typically triggers water inflow from the North Sea through the Danish straits and cause higher sea level in the Baltic Sea (Andersson, 2002; Bednorz and Tomczyk, 2021), thus favoring the occurrence

of compound events. Winter compound events at the individual tide gauges are mostly controlled by AO and NAO, which correlate significantly with all nine tide gauges (Fig. 8b). Furthermore, the correlation coefficients between the number of compound events and the average ACP indices are also generally higher in winter than in autumn, which indicates that the large-scale pattern controls the occurrence of compound events more effectively in winter than in autumn.

For the whole September-March period, the correlation between total number of compound events and the ACP indices was

found to be -0.64 with SCA, 0.5 with AO and 0.35 with NAO ($p < 0.05$, not shown). East Atlantic (EA), East Atlantic/Western Russia (EA/WR) and Polar/Eurasia (POL) patterns were found to have weaker and mostly non-significant correlations to the compound events (not shown), so they are not further considered in this study.

In Sect. 4.2 it was found that some tide gauges located in the Gulf of Bothnia (Kemi, Oulu and Rauma) have statistically significant ($p < 0.05$) increasing trends of the compound events during the 1961-2020 period. We analysed the trends of ACPs

over September-March period in 1961-2020 and found that while SCA has no trend, both AO and NAO have statistically significant increasing trends (determined by Mann-Kendall test, not shown). SCA pattern drives the compound events more in autumn while AO and NAO are more important in winter (Fig. 8). Thus, the increasing trends of compound events (Fig. 5a, b and e) can be probably linked to the increasing tendency of positive AO- and NAO-related circulation patterns during the last decades.

## 5 Discussion

### 5.1 Limitations of the analysis

Our precipitation analysis was based on FMI ClimGrid dataset (Aalto et al., 2016). We selected 70 km x 70 km area around the tide gauge and defined the daily precipitation sum as an average over this area. The caveat in this method is that small-scale precipitation extremes tend to be smoothed. Hence, the precipitation extremes captured in this study are mostly associated with

large-scale frontal precipitation. On the other hand, intense small-scale precipitation is often convective by its nature and thus mostly occur in summer when sea level is rarely high (Fig. 3). The actual precipitation values corresponding to the percentile thresholds in Table 2 are thus relatively small, compared to point-based observations.

We investigated the synoptic patterns behind the compound and non-compound events by presenting composite maps of various meteorological variables. This approach naturally hides the case-to-case variability between the events. Especially in

non-compound events, the samples behind the composites consist of 200-300 different cases (Fig. 6b-c and 7b-c), and hence the averaging smooths the variability of the synoptic patterns. Therefore, the composite maps need to be interpreted with caution.





## 5.2 Variability of compound events and the effects of large-scale circulation

Tide gauges located in the end of Gulf of Bothnia or Gulf of Finland tend to have their climatological peak of compound events earlier in the autumn than the other tide gauges (Fig. 4). When looking at extreme sea level events alone (Fig. 3, dashed lines),
Kemi and Oulu have more EL sea level events in September and October relative to the other tide gauges, and correspondingly less events in January-March period. This variability might be related to earlier freeze-up of the sea and the formation of land-fast ice in the ends of bays like the Gulf of Finland and the Bothnian Bay, which restrains the occurrence of the most extreme sea level heights later in the winter. In Kemi, the median date of the formation of seasonal ice cover is 2 December, while for instance in Helsinki, the date is 21 January (FMI, 2021).

The composite maps of meteorological variables indicate that the compound events are typically caused by closely-passing extratropical cyclones (ETCs). Although this finding was a fairly expected result, it has not been comprehensively documented earlier. Finland is a country located in the end of North Atlantic storm track (e.g. Wernli and Schwierz, 2006), and the strongest ETCs affecting northern Europe tend to originate and occur over the Norwegian Sea and the Barents Sea (Laurila et al., 2021). For this reason, the correlation between precipitation and sea levels is very modest on the Finnish coast (Fig. 2), and extreme
precipitation and extreme sea levels rarely occur coincidently. Nevertheless, high sea level together with high precipitation can cause notable impacts even if the variables are not extreme in isolation.

In annual basis, we found that the compound events are mostly driven by Scandinavian pattern in autumn, while Arctic Oscillation was found to be the most influential pattern in winter. When considering the total September-March period when compound events generally occur (Fig. 4), SCA pattern was found to dominate. North Atlantic Oscillation, which is often
considered the leading mode of variability in Euro-Atlantic region, was found to have only the 3rd strongest impact to the total number of compound events. The fundamental difference in these patterns is that SCA is the manifestation of highly meridional circulation while NAO reflects the zonal pattern.

Somewhat surprisingly, NAO was not the strongest driver of the compound events in autumn at any of the tide gauges and the correlations are mostly insignificant (Fig. 8a). In winter, NAO has clearly stronger influence. This suggests that while
compound events in autumn happen with variable phase of NAO, they tend to occur with positive NAO in winter. One notable aspect is also the fact that AO has higher correlation to the compound events than NAO. Although AO and NAO are highly correlated with each other, they have essential differences in the underlying mechanisms. AO reflects the zonally symmetric, annular mode of the whole Northern Hemisphere, while NAO represents more the local patterns in the Atlantic sector (e.g. Ambaum et al., 2001).

Our results are in line with the recent findings on the relationship of ACPs and the sea level variations in the Baltic Sea: Bednorz and Tomczyk (2021) found that SCA pattern was the most relevant of sea level variations on a monthly basis. They also documented that AO provides slightly stronger relationship with the sea level of Baltic Sea than NAO. In earlier studies it has also been documented that NAO controls the long-term sea level variability in the Baltic Sea (Andersson, 2002; Johansson et al., 2003; Dailidienė et al., 2006), but in these studies other ACPs were not considered.





Sea level extremes occur most often in winter season (Fig. 3), while extreme precipitation are the most frequent in summer and autumn (Fig. 3) and thus potentially influenced by different atmospheric circulation patterns than sea level extremes. Indeed, Irannezhad et al. (2014) studied the long-term precipitation variations and their relations to ACPs, and found that the most influential pattern on precipitation depends strongly on season. Based on their study, winter precipitation is mostly driven by NAO, but autumn precipitation is more controlled by SCA. Summer and spring precipitation are in turn most correlated

with EA/WR pattern. We found that NAO and AO mostly affect to the compound events in winter while SCA is the dominant mode of compound event variability in autumn (Fig. 8), which agrees well with the results by Irannezhad et al. (2014).

### 5.3    The relevance of compounding extreme precipitation and high sea level

The high sea level causes coastal flooding which is directly connected to the sea. Extreme precipitation instead induces flooding independent of overflow of sea level (i.e. pluvial flooding). Intense and long-lasting precipitation may also induce high river

discharge and consequently fluvial flooding, but this impact is limited to the vicinity of rivers and in addition, may not always materialize if the precipitation is not strong over the whole catchment area of the river. However, in urban areas with a notable share of sealed surfaces covering soils, such as the city centre of Helsinki, simultaneous extreme precipitation with high sea levels can amplify the impacts of coastal flooding by increasing overflowing sewers and slowing the drainage of water from low-lying sites such as basements and metro tunnels. On the other hand, pluvial flooding alone may also cause large damages,

as happened in the city of Pori at the delta of a flood prone river flowing into the Bothnian Sea (City of Pori, 2009). In the Finnish nuclear power plants on the shore of the Gulf of Finland and the Bothnian Sea, actions have been made to ensure that even in an unprecedentedly severe flooding event, neither seawater nor rainwater would flow onto the floors or foundations of buildings, thereby hampering the normal operation of the plant (TVO, 2008; Jänkälä, 2016).

In a very recent report by the Finnish Climate Change Panel, especially the future risk for pluvial flooding was estimated

to increase in the whole country (Gregow et al., 2021), and in Helsinki, flooding has been assessed to be one of the most significant weather-related risks to society (Pilli-Sihvola et al., 2018). Recommendations for minimum building elevations on the Finnish coast have been assessed in Kahma et al. (2014). These recommendations are based on the sea level in 2100 with an exceedance frequency of one event per 250 years, by taking into account the long-term and short-term variations of sea level in the future. Likewise, the most recent flood risk management plans for the Helsinki region (Jaakonaho et al., 2015) and

the whole country (Parjanne et al., 2018) focus mostly on the impacts of coastal flooding caused by extreme sea level and on river floods. Furthermore, urban stormwater runoff and management of pluvial flooding in Finland have been considered in several papers (e.g., Valtanen et al., 2014; Khadka et al., 2020). However, the combined effects from simultaneous high sea level and extreme precipitation have rarely been assessed. Even so, it has been recognized in the coastal city of Turku that simultaneously occurring seawater floods and heavy precipitation sometimes cause large-scale urban floods, with significant

concrete challenges to stormwater management (e.g., City of Turku, 2016; Gustafsson, 2018).

Finally, we emphasize that in this paper, we studied the atmospheric and marine conditions which create only the potential for compound flooding. Quantifying the actual compound flooding level is challenging. According to Bevacqua et al. (2020a), this is because river mouths typically do not have tide gauges (this is also the case in Finland), and in case of pluvial flooding, likely





due to scarcity of data. Nevertheless, Bevacqua et al. (2020a) presented that precipitation-based compound flooding analysis
can provide satisfactory information on the flooding potential in small- and medium-sized estuaries, with river catchment
smaller than about $5 - 10 \times 10^3$ km$^2$. In Finland, most of the rivers flowing into the Baltic Sea are relatively small. For this
reason, we believe that using precipitation as a proxy for compound flooding estimates provides reasonable results.

## 5.4  Future projections

In the warming climate, the days with extreme precipitation are expected to become more common in Finland (Lehtonen et al.,
2014; Lehtonen and Jylhä, 2019). The majority of the precipitation received in Finland is associated with ETCs (Hawcroft
et al., 2012), and it has been projected that the precipitation of ETCs will increase with the warming of the climate (Hawcroft
et al., 2018; Seneviratne et al., 2021). Therefore, even if the frequency of ETCs would not change significantly, one might
expect that the compound flooding events would become more common due to increased precipitation associated with these
storms. This is in line with our finding of increasing trends of the compound events in the Bothnian Bay, and we speculate
that also other parts of Finnish coast might see increasing frequency of these events in the future, in particular in the Gulf of
Finland where the long-term rise of sea level relative to the land is expected to be strongest (Pellikka et al., 2018).

## 6  Conclusions

In this paper, the potential for coastal flooding arising from joint occurrence of high precipitation and high sea level in Finland
has been investigated. Our first objective was to characterize the climatology, variability, and possible trends of the compound
events. While the sea level variations in the Baltic Sea are often quite pronounced, we found that the simultaneous occurrence of
high sea level with extreme precipitation on the Finnish coast is relatively rare: only around 5 % of all high sea level or extreme
precipitation cases are compounding events in which they both occur simultaneously. This is because the most extreme sea
levels take place typically in the winter while precipitation is the most abundant in the summer. Thus, compounding high
precipitation and high sea level events tend to occur between these seasons, during autumn and winter, with on average 98 %
of the EL compound events and 99 % HL compound events occurring during the cold season (September-March, Fig. 4).

Based on our results, the climatological frequency of EL (HL) compound events, averaged over 1961-2020 and across all
the nine tide gauges, is 2.1 (0.4) events per year. This means, for instance in Helsinki, that around twice a year, the sea level
exceeds 55 cm with at least 8.7 mm precipitation on the same day (Table 1). Respectively, around once per two years, the sea
level exceeds 70 cm with simultaneous precipitation of at least 13.1 mm. In addition, increasing trends of the compound evens
were found from Kemi, Oulu, and Rauma tide gauges (Fig. 5).

The second objective of this paper was to investigate qualitatively the synoptic patterns driving the compounding high
precipitation and high sea level events. By analysing composite charts of meteorological fields, we conclude that compound
events are typically associated with close passages of ETCs. Furthermore, the fingerprint of compound events in the sea level
pressure fields differed from the non-compound events in a way that high sea level events were associated with lower pressure
further away in the north, and high precipitation events were associated with only a weak or non-existing low pressure area.



Finally, as a third aim of this study, we analysed the different atmospheric circulation patterns favoring the compound events. In line with earlier studies considering only sea level (Bednorz and Tomczyk, 2021) or precipitation (Irannezhad et al., 2014) alone, negative Scandinavian pattern was found to have the strongest control to the occurrence of compound events. Negative SCA pattern favors ETC activity in the vicinity of Finland and thus induces the compounding high precipitation and high sea

level events.

This paper assessed climatological characteristics of joint heavy precipitation and high sea level events and the meteorological drivers leading to these compound events. Further research should focus on the impacts and the potential damages caused by the compound events, and quantify how important these events are for the risk of coastal flooding on the Finnish coast. This would be beneficial information for the coastal flood risk assessments in Finland now and in the future climate.

*Data availability.* Sea level observations used in this study are available from Jani Särkkä and Ulpu Leijala upon request. FMI ClimGrid data are available from https://paituli.csc.fi/download.html. ERA5 reanalysis data are available from Copernicus Climate Data Store: cds.climate.copernicus.eu. Atmospheric circulation pattern indices are available from https://www.cpc.ncep.noaa.gov/data/teledoc/telecontents.shtml.





## Appendix A: Appendix A

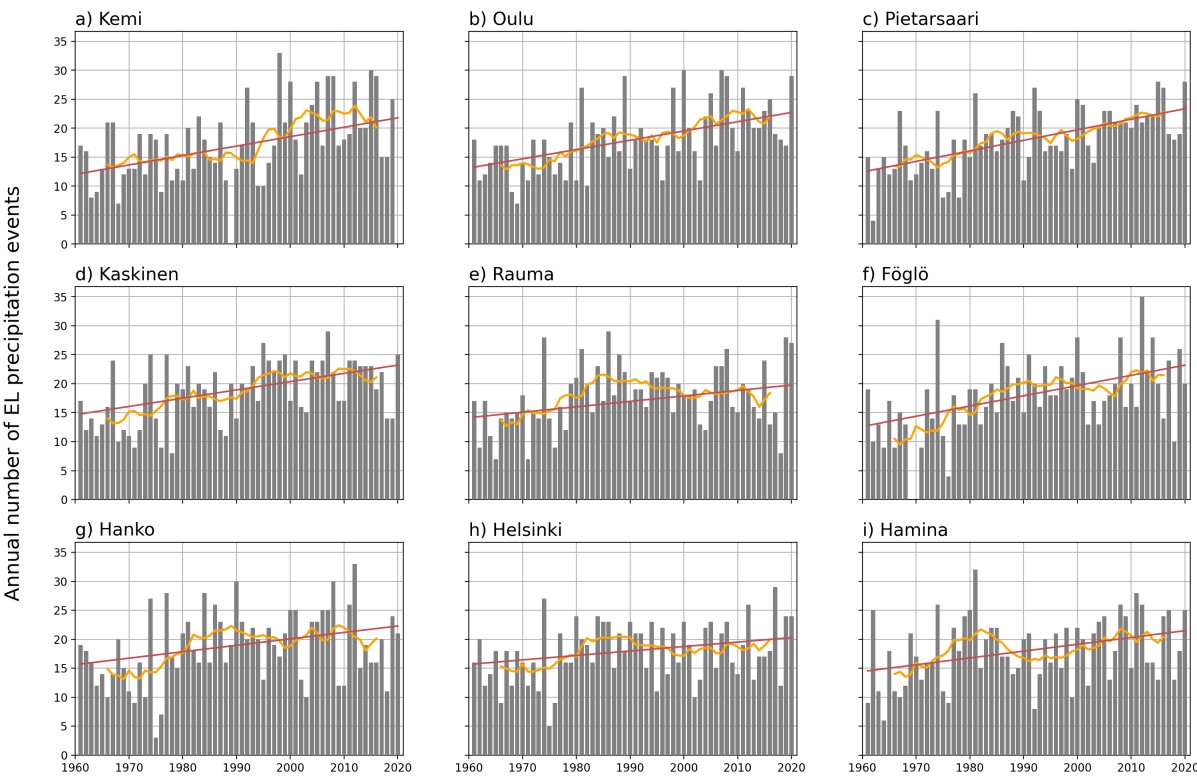

**Figure A1.** Annual number of single precipitation events at the tide gauges (grey bars), 10-year running mean of the annual values (orange line) and the estimates of their linear trends calculated with Theil-Sen method (red line). The linear trends are plotted only for those tide gauges whose Mann-Kendall trend estimate was increasing.





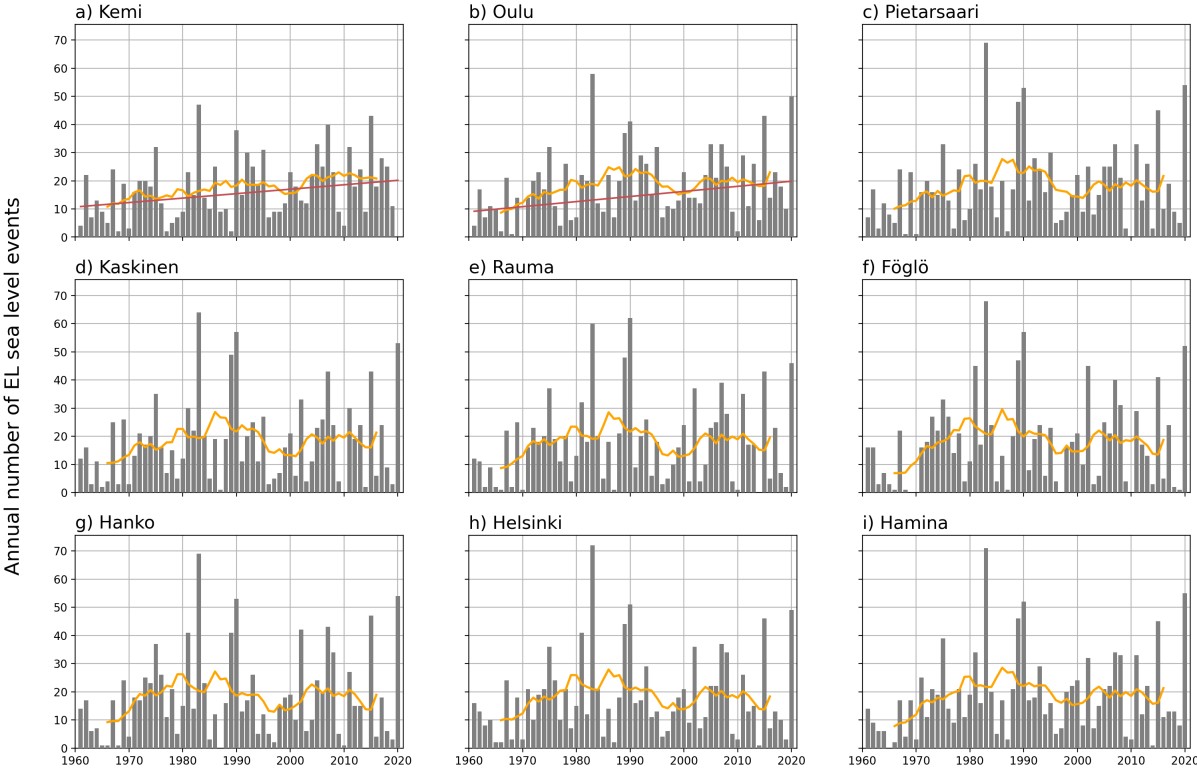

**Figure A2.** Same as Fig. A1 but for EL sea level events.

*Author contributions.* MR performed most of the data analysis and wrote the initial draft of the manuscript. The study was originally initiated by JR, together with KJ. UL and JS delivered the sea level observations. MR, KJ, JS and UL commented the manuscript and discussed the results at all stages.

*Competing interests.* The authors declare that they have no conflict of interest.

*Acknowledgements.* This research has been supported by the Finnish State Nuclear Waste Management Fund (grant nos. Dnro SAFIR
8/2019, 6/2020 and 6/2021). Copernicus is acknowledged for making ERA5 reanalysis available, and Climate Prediction Centre of National Oceanic and Atmospheric Administration is acknowledged for making the circulation indices available.



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
