# Peer review of "Characteristics of joint heavy precipitation and high sea level events on the Finnish coast in 1961-2020"

_Natural Hazards and Earth System Sciences, 2021_

## Referee Comment (RC1)

**Review of "Characteristics of joint heavy precipitation and high sea level events on the Finnish coast in 1961-2020"**

23 Nov. 2021

The co-occurrence of extreme precipitation and extreme sea levels aggravate the flooding impacts on coastal areas, thus compound flooding should be taken into account for a complete risk coastal assessment. In this study, a co-occurrence counting is used to quantify the compound effect of extreme precipitation and sea levels along the coast of Finland using observations.

The co-occurrence of heavily precipitation and extreme sea levels have been previously analyzed in Europe, including the coasts of Finland (Bevacqua et al., 2019), finding similar results as those showed in the present manuscript. However, Bevacqua et al (2019) included the wave contribution when defining extreme sea levels, which is a plus. They also used reanalysis and modelled data to represent both sea level and precipitation. Other than that paper, the compound effect of extreme sea level and precipitation along the Finnish coast has not been assessed. The novelty of the present manuscript is the use of observations for precipitation and sea level data. The authors have found low co-occurrence of extreme precipitation and sea level along the Finnish coast, which is itself a result. Findings show decadal variability in the compound effect as well as trends in some tide gauges.

However, although the results are promising, higher effort can be done to enrich the paper; the analysis is quite simple, and more information can be obtained from the data the authors show. For instance, the authors can show the return level of those individual extreme events (precipitation only and sea level only) in comparison with the return level when considering compound effect. This can be also performed with return periods (more details below). Regarding methodology, information is missing, and overall, the methodology needs further explanation and can be greatly improved (more details below). Also, I'm not sure if the correlation between compounding effect in precipitation and sea level can be associated with climate patterns; can be an arbitrary association resulting from the low number of compound events found (something to discuss with the authors). Another aspect is the organization of the paper; the introduction and conclusions are mixed up. Finally, I have found that the grammar and

language must be revised and notably improved before publishing. Therefore, I think the manuscript is not ready for publication; major revision.

**Suggested additions:**

- The co-occurrence of extreme events as counting of simultaneous threshold exceedance is one way of measuring compound effect. Other way that has been extensively used in the literature is the use of compound probability. Although this method is slightly more complicated to implement, it allows to answer some questions such as "what is the return level (or period) of an individual extreme event and how does it change when accounting for compound effect?" In addition, the results can be more easily comparable with previous works that have analyzed the joint return period of precipitation and extreme sea levels in the area (despite differences in the data used). A similar procedure can be found in Bevacqua et al (2019). Another way of enriching the paper would be performing a correlation analysis and significance testing, so the results are more robust. A similar method is used in Hendry et al (2019).

I truly believe that by implementing this type of analyzes, the authors can greatly improve the paper by enriching it with important information not only for the scientific community but also for stakeholders.

**Methods**

- When the authors define extreme sea levels and precipitation events (events above .95 or .98 thresholds), do they account for independency between them? If not, an overestimation of the co-occurrence of events could happen; two consecutive extreme sea levels and two consecutive extreme precipitation events driven by the same weather system should be counted as only one compound event.

- In the manuscript, the authors define co-occurrence as the threshold exceedances occurring simultaneously. They don't define what "simultaneously" mean (is it in the same day? In a 3-day time window?) Since the data is daily resolution, I'm assuming they calculate co-occurrence as the sea level and precipitation exceedances happening

in the same day. The authors may consider expanding this overlapping period to, for instance, ±1 or ±3 days (3 days was used in Bevaqcua et al., 2019). This will allow the authors to account for a weather system that caused precipitation one day and extreme sea levels the day after, for example. Also, it will allow to increase the sample size of extreme events co-occurrences, probably.

- Since the compound effect of precipitation and sea level has been assessed through a counting method, the authors couldn't assess the significance of the co-occurrence. I suggest to calculate the Kendall's rank correlation, which captures non-linear relationships, as performed in previous works (Hendy et al., 2019).

- Did the authors tested other time periods for the accumulated precipitation?

**Minor comments:**
- Line 4: in line 4 the authors define "compound events" but they already used this concept word in the line before. I would define it before using it.
- Lines 3 to 5 sound redundant.
- Line 23: It doesn't have to be anomalous to be a compound event.
- Lines 59 to 60: Why are the probabilities of coastal floods increasing? Is that the result of surges, waves, mean sea level rise, or a combination of all of them?
- Line 87: What kind of data is included in the reanalysis data?
- Line 91: "*In total there are 14 tide gauges on the Finnish coast, but only nine of them were used in this study*", why is that?
- Line 95*:* I would recall here that this is the highest value over 12 hours.
- Line 98: previous works have recommended not to use a linear trend to detrend sea level time series (Arns et al., 2013).
- Line 110: Why do you use another precipitation dataset near the borders? Also, I would indicate that FMIClimGrid are observations at the beginning of the paragraph.
- Line 111: "*but the number of stations has decreased towards the 21th century due to the automation of the measurement protocol*", this needs more explanation.

- Line 159: Do you consider a time window between threshold exceedances to assure independency?

- Line 167: How many of those compound events have happen in the same year? That could be relevant when calculating the correlation with climate indices.

- Line 166: *"When calculating the numerical values corresponding to the percentiles of precipitation, also days with no precipitation were taken into account"*, Why?

- Line 167: if HL represents events over a higher threshold, all the events included in HL should be also included in EL. Then, how is it that there are more events in HL than in EL?

- Line 170: The authors may want to allow a time window between extreme precipitation and extreme sea level to consider co-occurrence, taking into account the lag in the storm. This is, the same storm can cause an extreme precipitation one day, and extreme water level the day after. This probably will increase the number of co-occurrences.

- Line 185: have the authors calculated the trends on the extreme events alone or over the co-occurrences?

- Line 221: highest sea level variability doesn't imply higher sea level values.

- Line 275: Have the authors tested if the composite maps for compound events are statistically different from the composite maps of non-compound events? The composite maps of total column of water and 10-metre wind speed for sea level only and for compound look very similar. The fact that the number of observations is notably smaller in the compound composite map (N= 27) in comparison with sea level only (N= 221), could lead into differences between the composite maps. Thus, the differences showed in Figure 6 and Figure 7 between the compound and sea level only maps can be derived from the number of observations rather than from physical process.

- Line 320: One can argue that, with only 44 to 66 compound events (EL), and knowing a weak dependence between precipitation and extreme sea level in the region, the correlation between compound events and circulation patterns is arbitrary. Also, despite being statistically significant the correlation coefficients are generally small (Figure 8).

- Line 376: can you illustrate this idea with an example in Finland?
- Line 403: *"The high sea level causes coastal flooding which is directly connected to the sea"*, I believe this sentence is redundant.
- Line 404: Section 5.3n looks more Introduction to me.
- Line 424: "rarely" means "some", can you cite them? Have they found similar results?
- Lines 451 -455: these are results, are they mentioned in the Results section?

**References**

Bevacqua, E., Maraun, D., Vousdoukas, M. I., Voukouvalas, E., Vrac, M., Mentaschi, L., & Widmann, M. (2019). Higher probability of compound flooding from precipitation and storm surge in Europe under anthropogenic climate change. *Science advances*, *5*(9), eaaw5531.

Hendry, A., Haigh, I. D., Nicholls, R. J., Winter, H., Neal, R., Wahl, T., ... & Darby, S. E. (2019). Assessing the characteristics and drivers of compound flooding events around the UK coast. *Hydrology and Earth System Sciences*, *23*(7), 3117-3139.

---

## Author Comment (AC1)

**Response to Reviewer #1 of "Characteristics of joint heavy precipitation and high sea level events on the Finnish coast in 1961–2020"**

Mika Rantanen, Kirsti Jylhä, Jani Särkkä and Ulpu Leijala

We thank the reviewer for their constructive comments on our submitted manuscript. The point-by-point reply to the comments of the reviewer are below. Your comments are marked in black and our responses in blue.

**Reviewer #1**

**Review of "Characteristics of joint heavy precipitation and high sea level events on the Finnish coast in 1961-2020"**

**23 Nov. 2021**

The co-occurrence of extreme precipitation and extreme sea levels aggravate the flooding impacts on coastal areas, thus compound flooding should be taken into account for a complete risk coastal assessment. In this study, a co-occurrence counting is used to quantify the compound effect of extreme precipitation and sea levels along the coast of Finland using observations.

The co-occurrence of heavily precipitation and extreme sea levels have been previously analyzed in Europe, including the coasts of Finland (Bevacqua et al., 2019), finding similar results as those showed in the present manuscript. However, Bevacqua et al (2019) included the wave contribution when defining extreme sea levels, which is a plus. They also used reanalysis and modelled data to represent both sea level and precipitation. Other than that paper, the compound effect of extreme sea level and precipitation along the Finnish coast has not been assessed. The novelty of the present manuscript is the use of observations for precipitation and sea level data. The authors have found low co-occurrence of extreme precipitation and sea level along the Finnish coast, which is itself a result. Findings show decadal variability in the compound effect as well as trends in some tide gauges.

However, although the results are promising, higher effort can be done to enrich the paper; the analysis is quite simple, and more information can be obtained from the data the authors show. For instance, the authors can show the return level of those individual extreme events (precipitation only and sea level only) in comparison with the return level when considering compound effect. This can be also performed with return periods (more details below). Regarding methodology, information is missing,

and overall, the methodology needs further explanation and can be greatly improved (more details below). Also, I'm not sure if the correlation between compounding effect in precipitation and sea level can be associated with climate patterns; can be an arbitrary association resulting from the low number of compound events found (something to discuss with the authors). Another aspect is the organization of the paper; the introduction and conclusions are mixed up. Finally, I have found that the grammar and language must be revised and notably improved before publishing. Therefore, I think the manuscript is not ready for publication; major revision.

**Suggested additions:**

- The co-occurrence of extreme events as counting of simultaneous threshold exceedance is one way of measuring compound effect. Other way that has been extensively used in the literature is the use of compound probability. Although this method is slightly more complicated to implement, it allows to answer some questions such as "what is the return level (or period) of an individual extreme event and how does it change when accounting for compound effect?" In addition, the results can be more easily comparable with previous works that have analyzed the joint return period of precipitation and extreme sea levels in the area (despite differences in the data used). A similar procedure can be found in Bevacqua et al (2019). Another way of enriching the paper would be performing a correlation analysis and significance testing, so the results are more robust. A similar method is used in Hendry et al (2019).
I truly believe that by implementing this type of analyzes, the authors can greatly improve the paper by enriching it with important information not only for the scientific community but also for stakeholders.

Thank you for these useful comments. We agree that calculating the compound probabilities would be indeed a useful way to assess the importance of compound flooding events. However, although the joint probabilities of heavy precipitation and high sea level may be possible to quantify using for example copula-based analysis as done in Bevacqua et al. (2019), it would be a major effort that goes beyond what is possible with our resources for this manuscript.

Nevertheless, in order to answer the question "what is the return level (or period) of an individual extreme event and how does it change when accounting for compound effect" we conducted an extreme value analysis of the extreme precipitation and sea level events. We carried out the analyses separately for single events and compound events, and thus obtained information about how the probability of the events changes when the compound situations are considered.

The extreme events were extracted from the time series using the peaks-over-threshold (POT) method, and then a generalized pareto (GP) distribution was fitted to the extracted extreme events. We considered only the

September-March time period. An alternative of using block maxima and generalized extreme value (GEV) distribution proved to result in somewhat more uncertain results.

As an example of the results, at the Helsinki tide gauge, the return level of a 10-year heavy precipitation event is 30±3 mm, while for compounding events the return level is 23±3 mm. Thus, when taking into account the compound effect, the probability of the event clearly decreases.

The full documentation of the method, and the results are provided in the revised manuscript. As the extreme value analysis was somewhat unfamiliar for us, our colleague Mikko Laapas carried out the analyses. Thus, he was added as one of the co-authors in the revised manuscript.

**Methods**

- When the authors define extreme sea levels and precipitation events (events above .95 or .98 thresholds), do they account for independency between them? If not, an overestimation of the co-occurrence of events could happen; two consecutive extreme sea levels and two consecutive extreme precipitation events driven by the same weather system should be counted as only one compound event.

We took into account the independence of the events. This was stated in the manuscript at lines 181-184. In the cases when compound or single events took place on consecutive days, the largest in magnitude of them was only included in the analysis. Thus, the events have at least 48 hours time distance between each other. We believe that this is a sufficient time distance to consider the events as independent.

- In the manuscript, the authors define co-occurrence as the threshold exceedances occurring simultaneously. They don't define what "simultaneously" mean (is it in the same day? In a 3-day time window?) Since the data is daily resolution, I'm assuming they calculate co-occurrence as the sea level and precipitation exceedances happening in the same day. The authors may consider expanding this overlapping period to, for instance, ±1 or ±3 days (3 days was used in Bevaqcua et al., 2019). This will allow the authors to account for a weather system that caused precipitation one day and extreme sea levels the day after, for example. Also, it will allow to increase the sample size of extreme events co-occurrences, probably.

Thank you for this suggestion. We indeed defined the compound events as exceedances which occur on the same day. We agree that expanding the overlapping period to ±1 or ±3 days could be relevant when ensuring the independence of the events.

However, regarding the compound events, the closer they occur to each other, the more relevant they are impact-wise. For example, if the high sea level occurs two days after the heavy precipitation, the impacts of the event may not be amplified.

In addition to complicating the interpretation of the impacts, expanding the time window would also know subjective choices as to which of the consecutive days would be chosen as the actual time of the event.

it would also lead to subjective choices about which of the consecutive days to choose as the actual time of the event.

For these reasons, we decided to stick with the original simple definition of event co-occurrences, and we define the events to occur during the same day. Please note that we indeed take into account the independence of the events (see the previous response).

- Since the compound effect of precipitation and sea level has been assessed through a counting method, the authors couldn't assess the significance of the co-occurrence. I suggest to calculate the Kendall's rank correlation, which captures non-linear relationships, as performed in previous works (Hendy et al., 2019).

As suggested, Kendall rank correlation coefficients between the precipitation and maximum sea level have been added to Figure 1, similarly as in Hendry et al. (2019). Brief discussion on the rank correlations is provided in the results at line 278.

- Did the authors tested other time periods for the accumulated precipitation?

We did not test other time periods than the one-day (24-hour) time period. The precipitation on each day is accumulated over 06-06 UTC time period, which is the standard observation period for precipitation used by the Finnish Meteorological Institute.

**Minor comments:**

- Line 4: in line 4 the authors define "compound events" but they already used this concept word in the line before. I would define it before using it.

The word "compound event" is now defined at the first line of the abstract.

- Lines 3 to 5 sound redundant.

We rephrased the sentence on lines 3-5 to remove the redundancy.

- Line 23: It doesn't have to be anomalous to be a compound event.

The word "anomalous" has been replaced with "multiple".

- Lines 59 to 60: Why are the probabilities of coastal floods increasing? Is that the result of surges, waves, mean sea level rise, or a combination of all of them?

The main cause for the increasing flood probabilities is the mean sea level rise. On the Finnish coast it is counteracted by the land uplift, which is smallest on the south coast. There are no clear indications of increase in surges or waves in the future (Pellikka et al., 2018).

We added a sentence "Due to global sea level rise, ..." to line 59.

Pellikka et al., (2018):
https://www.sciencedirect.com/science/article/abs/pii/S0278434316302060

- Line 87: What kind of data is included in the reanalysis data?

The ERA5 reanalysis data includes meteorological fields which were used to study the synoptic situations during the compound events. We used instantaneous mean sea level pressure, total column water and 10-meter wind speed data at 0.5° horizontal resolution. This information is described in Section 2.3.

- Line 91: "*In total there are 14 tide gauges on the Finnish coast, but only nine of them were used in this study*", why is that?

The sea level of the neighboring tide gauges is highly correlated with each other. Thus, using all of the 14 tide gauges is not necessarily needed. We selected the nine tide gauges which we believe are representative of the sea level variations at the coastline, because they are located in different sub-basins of the Baltic Sea. In addition, one of the five tide gauges which were not used in this study is Porvoo tide gauge. It was established in 2014, so its period of record is very short.

- Line 95: I would recall here that this is the highest value over 12 hours.

The observation period used in both sea level and precipitation is 24 hours. This is now added to line 123.

- Line 98: previous works have recommended not to use a linear trend to detrend sea level time series (Arns et al., 2013).

We believe that this may concern mostly the global sea level time series. In Finland, the relative sea level rise (mean sea level rise minus land uplift) has proceeded mostly linearly, as the land uplift has exceeded mean sea level rise until the end of 20th century. In earlier studies, such as Johansson et al. (2014), these long-term trends have been removed simply by subtracting the linear trend. We follow this procedure. In the revised manuscript, we added a mention of the linear assumption to line 130.

- Line 110: Why do you use another precipitation dataset near the borders? Also, I would indicate that FMIClimGrid are observations at the beginning of the paragraph.

Station records from the surrounding countries, i.e., Sweden, Norway, Estonia, and Russia, were used by Aalto et al. (2016) to improve the quality of the FMIClimGrid dataset near border regions. Aalto et al. (2016) downloaded these records from the European Climate Assessment & Dataset (ECA&D) database.

In our study, we used only the FMIClimGrid dataset for precipitation data. To avoid further confusion, we have removed the sentence related to the European Climate Assessment & Dataset and shortened the paragraph (see below). We also added a note that FMIClimGrid is an observational dataset only and does not utilize a numerical weather prediction model as done in reanalysis datasets.

"For precipitation observations, we used a daily gridded dataset called FMIClimGrid (Aalto et al., 2016). Along with several other meteorological variables, FMIClimGrid contains precipitation in a 10 km x 10 km grid, spanning from 1961 to present. The gridding has been done with an interpolation method called kriging (Matheron, 1963; Goovaerts, 1999). As external predictors for the kriging, the elevation of the station, its relative altitude, and its proximity to the nearest sea and lakes were used. We emphasize that FMIClimGrid is an observational dataset only and does not utilize a numerical weather prediction model as done in reanalysis datasets. See more information of the dataset from Aalto et al., (2016)."

Aalto et al. (2016): New gridded daily climatology of Finland: Permutation-based uncertainty estimates and temporal trends in climate. https://doi.org/10.1002/2015JD024651.

- Line 111: *"but the number of stations has decreased towards the 21th century due to the automation of the measurement protocol",* this needs more explanation.

This sentence is related to the FMIClimGrid precipitation dataset. In Aalto et al. (2016) they say that the raw observations used for the creation of the gridded dataset were decreased towards the 21th century because of the automation of the measurement protocol.

To avoid further confusion between the original precipitation dataset and the data used in our study, we decided to remove these sentences from the revised manuscript.

- Line 159: Do you consider a time window between threshold exceedances to assure independency?

Yes. We applied an algorithm which seeks events which take place on consecutive calendar days. In the case of single events, only the calendar day which had the largest sea level or precipitation was included in the analysis. In the case of

compound events, the day with highest sea level was included in the analysis. However, compound events rarely took place on consecutive calendar days, so the effect of this declustering algorithm was negligible. Thus, the events examined in this study have at least 48 hour time distance with each other. Therefore, we believe the events can be considered as independent events with reasonably good approximation. This methodology is written in Section 3.1.

- Line 167: How many of those compound events have happen in the same year? That could be relevant when calculating the correlation with climate indices.

Figure 5 of the manuscript shows the number of compound events happening in the same year. The number ranges from 0 to 12.

- Line 166: *"When calculating the numerical values corresponding to the percentiles of precipitation, also days with no precipitation were taken into account"*, Why?

We calculated the percentile thresholds of maximum sea level from all calendar days. Therefore, for the sake of consistency, also the precipitation thresholds were calculated from all calendar days, including those with no precipitation.

- Line 167: if HL represents events over a higher threshold, all the events included in HL should be also included in EL. Then, how is it that there are more events in HL than in EL?

Indeed, HL events are also included in the EL events. However, the numerical values at line 167 represent the threshold values used to define EL and HL events, not the number of events. The numerical threshold for HL events is naturally higher than that for EL events. We state at line 204 that the group of EL events include also HL events.

- Line 170: The authors may want to allow a time window between extreme precipitation and extreme sea level to consider co-occurrence, taking into account the lag in the storm. This is, the same storm can cause an extreme precipitation one day, and extreme water level the day after. This probably will increase the number of co- occurrences.

Thank you for this suggestion. We indeed defined the compound events as exceedances which occur on the same day. We agree that taking into account the lag between the precipitation and sea level could possibly increase the number of events. However, for example Hendry et al. (2019) found the majority of the sites in the UK to have the maximum correlation between daily skew surge and daily river discharge at the 0-day lag (i.e. on the same day). Although we did not investigate the lagged correlations, selecting the same day is a reasonable choice. This is also in line with the paper by Bengtsson (2016) who also studied the extreme events occurring only on the same day.

In addition, as already stated in the 3rd response of this letter, the interpretation of the events would become more complicated. Allowing a lag would also mean that all the compound events and therefore all the analyses presented in the manuscript would have to be repeated. This is a major effort that we chose not to make at this point.

However, we added a note on this issue to paragraph 5.1 (Limitations of the analysis) so that the readers can take it into account.

Hendry et al. (2019): https://doi.org/10.5194/hess-23-3117-2019
Bengtsson (2016): https://doi.org/10.1002/hyp.10815

- Line 185: have the authors calculated the trends on the extreme events alone or over the co-occurrences?

We calculate the trends and their statistical significance for both extreme events alone and also for the compound events. For compound events, the statistically significant trends are shown in Fig. 5. For extreme events alone, they are shown in the Appendix figures A1 and A2.

- Line 221: highest sea level variability doesn't imply higher sea level values.

This is true. In principle higher variability can mean higher frequency but smaller amplitude. We removed the word "thus" from line 277.

- Line 275: Have the authors tested if the composite maps for compound events are statistically different from the composite maps of non-compound events? The composite maps of total column of water and 10-metre wind speed for sea level only and for compound look very similar. The fact that the number of observations is notably smaller in the compound composite map (N= 27) in comparison with sea level only (N= 221), could lead into differences between the composite maps. Thus, the differences showed in Figure 6 and Figure 7 between the compound and sea level only maps can be derived from the number of observations rather than from physical process.

We used a two-sided T-test to calculate whether or not the samples of compound events and non-compound events result in identical maps. We added stippling to the maps to indicate where the averages across the non-compound events are statistically different from those of the compound events.

Furthermore, we also use compound and non-compound events from the nearby stations to increase the sample sizes used to produce the composite maps. Thus, for Kemi composite, we also use events from the Oulu tide gauge (Fig. 7 in the revised manuscript). For Hamina composite, we also use events from the Helsinki tide gauge (Fig. 8). Duplicate events were naturally neglected.

- Line 320: One can argue that, with only 44 to 66 compound events (EL), and knowing a weak dependence between precipitation and extreme sea level in the region, the correlation between compound events and circulation patterns is arbitrary. Also, despite being statistically significant the correlation coefficients are generally small (Figure 8).

The total number of EL compound events at the tide gauges varied actually from 97 (Pietarsaari) to 165 (Kaskinen and Kemi). The annual numbers are shown in Figure 5. Howerer, the reviewer still makes a valid point. The number of events reflects naturally the threshold used to define the event: the higher is the threshold, the fewer events there are. The choice of the threshold is somewhat subjective; for EL we wanted to have a reasonably low threshold to ensure enough events.

The annual occurrence of the events is not normally distributed, but the distributions resemble more Poisson distribution. For this reason, we used the Spearman rank correlation because it is nonparametric and does not assume normal distribution in the datasets.

In the revised manuscript, we mention that the correlation coefficients are generally small. Even so, the results are in line with the previous findings on atmospheric circulation patterns. Thus, we think that the correlation coefficients still reflect the real physical connection between the events and the atmospheric circulation.

- Line 376: can you illustrate this idea with an example in Finland?

We recognize that there are very few documented examples from Finland. In the stormwater management plan of the City of Turku, they mention pluvial flooding together with high sea level (coastal flooding) as one of the concrete challenges in the stormwater management.

We rephrased the sentence to "Nevertheless, high sea level together with high precipitation might cause notable impacts even if the variables are not particularly extreme in isolation."

- Line 403: *"The high sea level causes coastal flooding which is directly connected to the sea"*, I believe this sentence is redundant.

This sentence has been removed.

- Line 404: Section 5.3n looks more Introduction to me.

We moved Section 5.3 to Introduction with some small modifications to the text.

- Line 424: "rarely" means "some", can you cite them? Have they found similar results?

The word "rarely" has been removed. The sentence reads now "However, to our knowledge, the combined effects from simultaneous high sea level and extreme precipitation have not been assessed in Finland."

- Lines 451 -455: these are results, are they mentioned in the Results section?

We added more analysis on the return levels of the events to the conclusions. Thus, this part of the conclusions was removed.

---

## Author Comment (AC2)

**Response to Reviewer #2 of "Characteristics of joint heavy precipitation and high sea level events on the Finnish coast in 1961–2020"**

Mika Rantanen, Kirsti Jylhä, Jani Särkkä, Mikko Laapas and Ulpu Leijala

We thank the reviewer for constructive comments of our submitted manuscript. The point-by-point reply to the comments of the reviewer are below. Your comments are marked in black and our responses in blue.

This work presents a joint analysis between sea level and precipitation observations around the coastlines of Finland to investigate the occurrence of compound events that are relevant for coastal flooding episodes. The results are generally well described and presented, although the analyses are sometimes superficial. I am giving some suggestions below, followed by a list of minor issues that I have found in the manuscript. I believe that this work has the potential to be published, after the points below have been addressed.

Section 4.2: why is the timing of the compound events different between tide gauges in the end of the gulfs and those closer to the Baltic? what are the mechanisms explaining this difference? is it because extreme sea levels are caused by different processes in the two areas?

We speculate that the slightly different timing of the compound events is related to the earlier freeze-up of the sea in the end of the Gulf of Finland and the Bothnian Bay, which restrains the occurrence of the most extreme sea level heights later in the winter. We discussed this finding in the first paragraph of Section 5.1.

Section 4.3: The idea of the composite maps to investigate the patterns that lead to compound and non-compound events is interesting. But the robustness of the results shown in the maps in figures 6 and 7 should be discussed. Only 27 and 28 situations are used to produce these maps and there is a lack of information about their similarities. I think deviations from the mean fields are needed to improve the interpretation of these results. In addition, to increase the size of the sample I suggest redoing these maps using various nearby tide gauges, instead of only one. This limitation is noted later in section 5.1, but without further development.

Thank you for this comment. For the revised manuscript, we have developed further figures 6 and 7, and the related discussion on them as follows:

- We added stippling to the figures to indicate where the difference of the non-compound mean field to the compound mean field is statistically significant. For this reason, we added sea level pressure to a separate row.

Thus, the figures 7 & 8 now have three rows (sea level pressure, total column water and 10-meter wind speed).
- We now also use nearby stations in the composites. Thus, for Kemi composite, we also use events from the Oulu tide gauge (Fig. 7 in the revised manuscript). For Hamina composite, we also use events from the Helsinki tide gauge (Fig. 8). Duplicate events were naturally neglected.
- We use only the months from September to March when compound events predominantly take place
- Deviations from the mean fields (i.e. anomaly charts) are added to the appendix.

We hope that the information given by figures 7 and 8 makes more sense now.

There is no information on the predominant month when these situations take place. I guess, from the seasonal cycles in figure 4, that precipitation extremes will dominate in summer and sea level in winter. I wonder if the synoptic patterns of precipitation-only and sea level-only extremes during the fall/winter season are different from those shown in figures 6 and 7, when all seasons are included. I think it would be more interesting to select the season when coincidences are possible; otherwise, what we are seeing is only the dominant pattern in different seasons.

As stated already in the previous response, we now use the September-March period in the composite figures. Thus, this issue is no longer present.

67: please, rewrite the sentence "that have been investigated on the Finnish coast not until the recent decade" as it is hardly understandable.

We rewrote the sentence to "that have been investigated on the Finnish coast in the recent decade".

100: on the use of the maximum daily sea levels, is there any difference due to different samplings before and after 1971?

We acknowledge that observations before 1971 are likely biased towards lower values due to coarser samplings. This issue is added to the revised manuscript to line 126: "Due to coarser sampling before 1971, the daily maximum values in the pre-1971 period may be slightly biased towards lower values."

111: 21tg to 21st

We chose to remove this sentence due to a confusion raised by another reviewer.

142: the term "orthogonal" can be misleading in this context. If calculated with rotational EOFs over monthly data, NAO, SCA and other patterns are by construction orthogonal to each other (on a monthly basis, not on a seasonal or yearly basis). The meaning of orthogonal here seems to refer to the spatial structure of the MSLP right?

Indeed, this is what was meant to communicate to the readers. We rephrased the sentence to "The spatial structure of MSLP in SCA pattern is orthogonal to NAO (i.e. the NAO pattern is zonal and the SCA pattern is meridional)."

170: change were to was

Thank you for pointing this out.

214: change are to is

Thank you for pointing this out.

Figures 6, 7: units are missing from the legends. Also in the caption, what does "average sea level height" means? Is it the average for the particular tide gauge during the selected episodes?

We decided to remove the legends regarding average sea level height and precipitation, because in the revised versions of the figures we also use the nearby tide gauges. Thus, it is not unambiguous how to define the average sea level or precipitation from events occurring in more than one tide gauges.

Section 5.3: this section seems to me more part of the introduction. Just a suggestion…

The section 5.3 was moved to Introduction with some small modifications to the text.

---

## Author Comment (AC3)

**Response to Reviewer #3 of "Characteristics of joint heavy precipitation and high sea level events on the Finnish coast in 1961–2020"**

Mika Rantanen, Kirsti Jylhä, Jani Särkkä, Mikko Laapas and Ulpu Leijala

We thank the reviewer for constructive comments of our submitted manuscript. The point-by-point reply to the comments of the reviewer are below. Your comments are marked in black and our responses in blue.

General comments:

The manuscript deals with the concurrent occurrence of heavy rainfall and high water events along the Finish coast. The paper is well written, and what is presented is sound. The only thing I do not understand is, what is the point the authors want to make.

If they wanted to point out that these compound events create more severe flooding then in the case of non-compound events, well, in this case the authors failed miserably. There is no indication, and no statistics shown that can convince me that it is important to look at compound events in order to understand the severity of these events.

Actually, I am not sure why the authors have written this paper. The atmospheric circulation might be interesting, but me as an oceanographer am always looking at implications that some variables have for storm surge. I do not think that you need to analyze rain data in order to find out about flooding events.

Maybe I am missing something, maybe I was expecting too much. But as it stands now, this article is not very interesting. On these points I do reject the manuscript, but the editor might see it differently.

Specific comments:

Abstract: the authors want to correlate the rainfall and water level to compute compound flooding. Already in the abstract, they state that these events mostly are due to sea level variability (nothing surprising). I am not sure how the rainfall fits into this picture…

Thank you for this comment. We would like to emphasize that in many previous papers the potential for coastal compound flooding has been quantified using precipitation and sea level data (e.g. Bevacqua et al. 2020a, 2019). Furthermore, as

stated in Bevacqua et al. (2020b), for small- or medium-sized rivers precipitation can be used as a proxy for river (fluvial) flood and thereby also for coastal compound flooding in those rivers. Thus, we think that using precipitation is a scientifically reasonable choice.

The correlation between precipitation and sea level was found to be rather modest in the Finnish coast. Even so, we believe the modest correlation is a result which is worth documenting. Our findings are thus in line with Bevacqua et al. (2019, their Fig. 2) who documented that the probability of compound flooding is relatively small in Finland compared to elsewhere in Europe.

Bevacqua et al. (2020a): https://doi.org/10.1038/s43247-020-00044-z

Bevacqua et al. (2020b): https://nhess.copernicus.org/articles/20/1765/2020/nhess-20-1765-2020.html

Bevacqua et al. (2019): https://www.science.org/doi/10.1126/sciadv.aaw5531

36-42: for this please see also Ferrarin et al, 2021 (Progress in Oceanography)

Thank you for pointing out this interesting and also very comprehensive study. We added a citation to this study at line XXX.

30: are you sure it is the precipitation, and not some other correlated variable like wind or atmospheric pressure that makes these events compound events?

The underlying reason for the weak correlation is the passing extratropical cyclones, so it is very likely that atmospheric pressure (negatively) correlates with the sea level height as well.

98: how strong is this trend (numbers)?

The magnitude of the negative trends that were removed range from -6.4 cm per decade in Pietarsaari to -1.0 cm per decade in Hamina. This information has been added to the revised manuscript at lines 131-132.

105: please make clear if these data are observations or come from a meteorological model.

Thank you. We added a sentence "We emphasize that FMIClimGrid is an observational dataset only and does not utilize a numerical weather prediction model, similarly as done in reanalysis datasets." to lines 142-143.

Table 1: have all points a MSL of 0?

MSL is not zero, MSL is given by annual mean sea level in N2000 reference height subtracted by the linear trend of the annual means.For some years and for some tide gauges MSL is close to zero but not exactly.

123: I guess there is also rain data in ERA5. Did you use it and compare it to the observation data? If not, why not? If yes, how did the two data sets compare?

Thank you for this comment. There is indeed total precipitation data available in ERA5, but we decided to not use it in our study. There are two main reasons for this choice:

1) We wanted to rely purely on observational data. This is one aspect of how our study differs from previous studies on compound flooding which have used global reanalysis datasets (for example Bevacqua et al. 2019). Thus, the use of observational datasets only can be considered as one novelty of our paper.
2) Furthermore, no direct observations of precipitation are assimilated to ERA5 over the pre-1979 period. Furthermore, Bell et al. (2021) state that the precipitation in ERA5 performs more poorly in Europe than in US, and given the coarser horizontal resolution in ERA5 (31 km) than in FMI ClimGrid (10 km), we do not specifically expect that ERA5 would perform better than FMI ClimGrid.

Bevacqua et al. (2019): https://www.science.org/doi/10.1126/sciadv.aaw5531

Bell et al. (2021): https://rmets.onlinelibrary.wiley.com/doi/full/10.1002/qj.4174

307-9: this is what I was fearing… that there is little correlation between rain and sea level. And for the whole article you are basically repeating it…

The correlation between precipitation and sea level is indeed rather modest, but in line with the previous studies. For example, Hendry et al. (2019) reported Kendall rank correlations between 0 and 0.3 for the precipitation and river discharge in the UK.

We added Kendall rank correlations to Figure 1 to emphasize more clearly the correlations between precipitation and sea level at the tide gauges.

375-6: Well, I was eager to see the prove of this statement, but I guess the authors do not convince me

We agree that this statement in question was perhaps a bit too exaggerated. We have rephrased the sentence to "Nevertheless, high sea level together with high precipitation might cause notable impacts even if the variables are not particularly extreme in isolation."

406-9: This is all hypothetical. Where can I verify this hypothesis?

Thank you for this comment. The storm water management plans of Turku and Helsinki both mention that high sea level simultaneously with intense rainfall may hamper drainage operations. For example, in Helsinki, the ends of sewers will have to be closed if the sea level is too high. If it rains heavily in these situations, the sewer system will easily start flooding the streets and basements of the buildings. This was the case for example in October 2021 (news article in Finnish): https://www.hs.fi/kaupunki/art-2000008349729.html.

However, the importance of these situations is poorly studied in Finland.

431-2: It is maybe a proxy for compound flooding, but I am not sure if this is important, because you did not show the importance of compound flooding.

We would like to emphasize that the main scope of our study was more to characterize the synoptic and climatological situations when these compound flooding events take place. This study acts as a first step in paving the way for further analysis on compound flooding and their direct importance for society.

However, as other referees also asked to present the likelihood of the events, we added a section which presents the probability of single events compared with compound events. This can be found from Section 4.3 from the revised manuscript.

The 443-450: Well, you should have really shown that sea level is higher statistically during compound flooding then without compound flooding. I didn't see this analysis, hopefully I didn't miss it.

We are not sure if we understood this comment correctly. We agree that the sentence "the potential for coastal flooding arising from joint occurrence..." is misleading because the coastal flooding is indeed caused by high sea level and not from precipitation.

We changed the wording to "the potential for flooding arising from joint occurrence...".

451-5: are these levels statistically different from non-compound events?

In the revised manuscript, we present more explicitly the return levels of the single events compared with the return levels of compound events. Thus, this part of conclusions was changed.